# EEVEE AND GATE: FINDING THE RIGHT BENCHMARKS AND HOW TO RUN THEM SEAMLESSLY

## ABSTRACT

Model evaluation is a cornerstone of machine learning, guiding model design and progress measurement. Designing generalizable evaluation processes remains a challenge, however, partly due to the vast number of possible domain, task and modality combinations and lack of knowledge of how informative they are. In this paper, we propose *EEVEE* (Efficient Evaluation process Evolution Engine)[1], a method that frames evaluation process design as a learning problem. By analyzing a large number of evaluation metrics from diverse benchmarks and models, EEVEE identifies a smaller subset of tasks with high predictive power over the full set of evaluation metrics, reducing evaluation time. To find the optimal subset maximizing signal while minimizing GPU hours, EEVEE evaluates pre-trained models of various architectures, pretraining schemes, and modalities on diverse downstream tasks and datasets including image classification, segmentation, relational reasoning, zero-shot image-to-text tasks, medical classification and segmentation, video classification, and regression. Our results identify three subsets of benchmarks, with 8, 15 and 21 tasks, providing high quality signal for model generalization. Key benchmarks selected include iWildCam, CLEVR-Math, ACDC, WinoGround, CIFAR100, Fungi, and ADE20K. We structure the subsets into three tiers for 12, 24, and 36 GPU-hour budgets and package them into a unified, efficient, and user-friendly Python framework that we built with the researcher in mind – which we refer to as the GATE engine. Our experiments reveal ConvNextV2, SigLIP and CLIP as top-performing model encoders, with EfficientNetV2 and ResNext50 excelling in medical tasks and challenging image classification, in particular in Happy Whale Individual classification, ConvNet based models seem to outperform transformer models by a factor of 2.5x, which is surprising. The top performing encoder being ConvNextV2 followed by CLIP seems to agree with other recent large scale evaluations. We also demonstrate the framework's versatility in fine-tuning models from text and audio modalities, paving the way for future cross-modal evaluations.

## 1 INTRODUCTION

**Increasing Complexities of Benchmarking:** As we create benchmarks for expanding model capability evaluation, the growing number and complexity of these benchmarks inadvertently complicates evaluation, requiring more resources like engineering, computation, and research time. Consequently, prioritizing which benchmarks to use becomes challenging. The high costs and longer wait times of newer, complex benchmarks often deter their adoption, leading researchers to rely on older, simpler benchmarks. This risks missing valuable insights from innovative ideas that may underperform on simpler benchmarks but have broader applicability, while promoting incremental improvements that overfit to simpler benchmarks but underperform in comprehensive evaluations.

To illustrate the mounting increase in available benchmarks, we can look at the historical benchmarks in deep learning. Few benchmarks have had as much impact as ImageNet (30), which remains a rich resource for model training and evaluation, particularly in visuo-linguistic models. As key capabilities for deep neural networks were discovered, more benchmarks were generated to measure and stimulate progress in those areas. In natural language processing, the GLUE benchmark (69),

---

[1]Pronounced as /ˈiːviː/ EE-vee

SQuAD (48), and CoNLL-2003 (51) have been instrumental. In audio processing, LibriSpeech (42), TIMIT (16), and VCTK (72) are widely used. For machine translation, WMT (3), IWSLT (23), and Europarl (26) have driven advancements. Relational reasoning has been advanced by benchmarks such as CLEVR (24), bAbI (70), and RAVEN (75). In segmentation, PASCAL VOC (14), Cityscapes (8), and COCO (36) remain crucial. Large language models are often evaluated using benchmarks like SuperGLUE (68), LAMBADA (43), and MMLU (20). Vision-language models are typically evaluated using benchmarks such as VQA (1), Visual7W (81), and Flickr30k (45).

As a result, a researcher has to choose from all these options, and even more, and then find a way to unify and experiment with their models across all of them. The lack of unification, and the lack of guarantees for their generalization signal, quickly becomes a kind of "evaluation hell", where researchers waste a lot of time just doing redundant things like fixing the same bugs to download datasets, preprocess them etc, while at the same time not having any real signal as to which benchmarks are more informative, other than just knowing what has been used the most – which is usually a function of popularity, and not real informativeness. To elaborate, the adoption of complex evaluation processes that could enhance research efficiency and impact is often hindered by the engineering effort required to evaluate machine learning models. Researchers must create involved pipelines across multiple datasets demanding high data engineering efforts, develop task-specific adapters, and derive nuanced training recipes, which is time-consuming. As a result, researchers often revert to simpler evaluation strategies instead of comprehensive assessments.

A good benchmark should alleviate these burdens by automating dataset handling, integrating task adapters, optimizers, schedulers, and logging mechanisms seamlessly. It should provide broad and meaningful signals with minimal GPU time, accommodating various computational budgets, ensuring inclusivity. Furthermore, an increasingly important factor for a robust modern benchmark engine is its support for multi-modal learning and early fusion techniques. AI systems must seamlessly integrate and reason across multiple modalities, such as text, images, audio, and more. Multi-modal learning enhances self-supervised learning opportunities and provides inherent supervision through natural alignments, like audio-visual synchronization in videos. Early fusion, where data from different modalities is combined at the initial stages of processing, allows models to leverage shared representations, improving generalization and reasoning capabilities across varied tasks and domains. These key desiderata are what motivates the production of this work.

With the desiderata in mind, we next introduce EEVEE, a methodology developed for building high-signal low-cost evaluation routines, and GATE, the resulting benchmark that is designed to be extensible, readable, flexible, modular and robust, supported by a new efficient, easy to use framework.

**EEVEE, Learning Optimal Benchmarks:** The ability to find which benchmarks offer the most signal with respect to a given goal, such that we can optimize our compute time, research iteration speed, and engineering time is increasingly crucial. In this work, rather than just manually designing a new set of benchmarks, we propose a methodology, called *EEVEE (Empirical Evaluation process Evolution Engine)* that frames evaluation design as a learning problem and then leverages machine learning to automate the discovery and refinement of evaluation processes.

More specifically, EEVEE operates by taking in a large set of performance metrics from diverse models applied across various benchmarks and identifies a smaller subset of benchmarks with high predictive power over the entire set. EEVEE achieves this through two main components: (a) an evolutionary algorithm to optimize the selection of benchmark combinations based on a computed score, and (b) a meta-model trained to predict a model's performance on the full set of benchmarks using performance metrics from a chosen subset. We parameterize the meta-model as as a small neural network.

The meta-model receives input performance metrics from a subset of benchmarks and predicts performance on the full set of performance metrics. Through careful $k$-fold cross-validation and leveraging a diverse set of models and benchmarks, EEVEE iteratively evolves benchmark combinations that offer high information content with respect to the entire spectrum of benchmarks, ensuring robust, efficient and comprehensive evaluation that can be targeted to computational budgets ranging from more "GPU Poor" users to high-budget organizations.

Taking the desiderata explained above and the resulting understanding of what a good evaluation engine should look like, we demonstrate the effectiveness of EEVEE by tasking it with the discovery

of benchmark combinations that offer good **signal-to-GPU-time** ratio, for the evaluation of **model encoders** – also referred to as backbones, on their ability to adapt to new tasks, domains, and modalities. For this purpose, we choose a pool of 20 models, varying in their pretraining schemes (e.g CLIP, DINO, ImageNet Classification), architectures (e.g. ResNets, ViTs, ConvNext) and even their source modalities (e.g. Whisper, BERT), which we adapt on 31 benchmarks ranging from image classification, segmentation, relational reasoning, zero-shot image-to-text tasks, medical classification and segmentation, video classification, and regression, using robust fine tuning recipes, and training for 10K iterations, ensuring that the signal we get is about models that are adaptable, generalizable and efficient in their adaptation.

By applying 20 models on 31 benchmarks and employing EEVEE on their resulting metrics, we identify three subsets of benchmarks, each targeted to a specific computational budget range. Some of the key benchmarks that have been selected include iWildCam, CLEVR-Math, ACDC, WinoGround, mini-ImageNet, Fungi, ADE20K, and dtextures. We refer to the discovered subsets as *Tiers*, and assign to them identifiers for their sizes, specifically, *small* (n=8, 12 GPU hours), *base* (n=15, 24 GPU hours) and *big* (n=21, 36 GPU hours). We package these tiers into our comprehensive benchmarking suite and software framework (called *GATE*) designed for domain, task and modality transferability evaluation, which facilitates the transfer of neural network encoders to different modalities, domains, and tasks. GATE's architecture caters to the research community, enabling straightforward replacement of these transferable encoders with minimal effort. With these innovations, GATE seeks to evolve the landscape of model encoder evaluation, championing a deeper understanding of transfer learning and model adaptability.

**Contributions:** 1. We introduce *EEVEE*, a machine learning approach for selecting subsets of benchmarks optimized to offer maximal predictive power over a larger benchmark set. 2. We conduct a comprehensive investigation of diverse benchmarks within the space of image, image+text and video modalities, pinpointing those with the highest predictive value for a model's performance in downstream tasks. We apply EEVEE to model encoder evaluation by training 20 models on 31 benchmarks, identifying subsets of 8, 15 and 21 benchmarks that offer high signal-to-GPU-hour ratios. 3. We pack the EEVEE-discovered subsets (of 8, 15 and 21 benchmarks out of 31 benchmarks) into targeted benchmark packs, referred to as tiers, designed for specific compute budgets (of 12, 24 and 36 GPU hours) and project phases, and establish standard experimental settings for these tiers. We call these collectively as the GATE Benchmarks. 4. We develop the *GATE* engine, a unified benchmark suite and software framework that automates dataset downloading, preprocessing, and pipelining for fine tuning and evaluation. GATE facilitates the incorporation of new model encoders, adapts input modalities, fine-tunes with robust recipes, and logs critical information such as training and evaluation metrics, power, energy, computational usage, task visualizations, and model gradients per layer. 5. Through our extensive investigation, we identify foundation models demonstrating superior transferability across diverse tasks. 6. We run a range of modality-shifting transfer experiments in the standard evaluation process for ML researchers, so that future work can potentially further probe into how pretraining on one set of modalities transfers to downstream (and potentially different) modalities.

## 2    RELATED WORK

**On the Diversity of Benchmarks:** There is a vast array of benchmark suites in machine learning. To the best of our knowledge, the benchmark suites relating strongly to GATE are ImageNet (9), VTAB (74), VLMBench (78) and WILDS (27). ImageNet has been of tremendous importance and interest to the transfer learning community. Nevertheless, there has been skepticism about overfitting to such datasets resulting from implicitly qualifying models using the test set performance over the years (49; 6) or the test set not being challenging enough to gauge model generalization power (50). Although ImageNet pre-training helps transfer performance to the many-shot classification setting (13), it provides minimal to no gains on more challenging datasets such as fine-grained classification (28). Similarly, with a larger distribution shift, ImageNet pre-trained models was found to offer limited benefits for medical imaging tasks due to large distribution shifts induced by fundamental differences in data sizes, features, and task specifications; that is, lightweight models perform comparably to standard architectures (47). To make matters worse, ImageNet performance is less correlated with and less predictive of downstream performance on diverse tasks beyond classification such as object detection, few-shot classification, and segmentation (13). On top of it all,

| Desiderata ↓ Benchmark → | ImageNet | VTAB | VLMBench | WILDS | GATE (Ours) |
|---|---|---|---|---|---|
| Diversity of Tasks | ✓ | ✓✓ | ✓✓ | ✓✓✓ | ✓✓✓ |
| Diversity of Domains | ✓✓ | ✓✓ | ✓✓ | ✓✓✓ | ✓✓✓ |
| Diversity of Modalities | ✓ | ✓✓ | ✓✓ | ✓✓✓ | ✓✓✓ |
| Automatic Dataset Download/Preparation | ✓ | ✓ | ✓ | ✓ | ✓✓✓ |
| Code allows for easy switch of encoders | ✓ | ✓ | ✓ | ✓ | ✓✓✓ |
| Optimized for fast and effective research iteration | ✓ | ✓ | ✓ | ✓ | ✓✓✓ |
| Run Time | ✓✓ | ✓✓✓ | ✓✓✓ | ✓✓ | ✓✓✓ |
| Includes Medical Domains | ✓ | ✓ | | ✓✓ | ✓✓✓ |
| Includes Environmental domains | ✓✓ | ✓✓ | ✓ | ✓✓✓ | ✓✓✓ |
| Tiered compute budgets | ✓ | ✓ | ✓ | ✓ | ✓✓✓ |
| GPU poor optimized | ✓✓ | ✓✓ | ✓✓ | ✓✓ | ✓✓✓ |

Table 1: Our Desiderata (first column) VS Benchmarks (first row). More ticks means better, from one/red/lacking, two/gray/OK, three/green/good

when ImageNet is extended with a perturbed temporal dimension, models performance significantly worsen (55).

**On the Usability of Benchmarks:** Beyond ImageNet, VTAB introduced a benchmark with a wider diversity of tasks and domains (74). Nevertheless, it does not offer task and domain shifts offered in GATE, such as medical segmentation and video classification and regression that are known to be ill-measured and gauged by ImageNet alone (47; 55). That said, VTAB offers satellite imaging and 3D tasks which GATE does not. Nevertheless, GATE as a software framework was optimized to minimise usage friction, to take no more than 12 GPU hours on our smallest tier, and, to only require approximately 1 hour of adding the new encoder and wrapping it into GATE wrappers for GATE to be able to go away and take care of everything, including dataset downloading, task adapter integration and full train/val and test cycles with logging of various key metrics. VTAB, in our experience, requires a lot more manual work in getting the datasets, and integrating new models to be adapted. Similarly, VLMBench (78) and WILDS (27) offer more diverse datasets beyond previous work but neither offer a tiered approach that enables iterative development of models during pre-training, nor produce extensible and flexible benchmarks that can be easily glued into researchers experimentation code without friction.

**On the Systematic Selection of Benchmarks:** Previous work investigated the properties inherit in multi-task benchmarks that trade-off diversity and sensitivity where the latter is how robust a benchmark ranking is to the inclusion of irrelevant models or minute changes in the tasks themselves (76). It was found that multi-task benchmark are unstable to irrelevant changes in tasks design. Nevertheless, this is related to how the benchmark ranks models; whether it compares how model often ranks higher than another in cardinal benchmarks or if the performance across tasks is averaged to produce a single rank in cardinal ones. Meanwhile, our benchmark produces fine-grained information to model performances across diverse tasks rather than producing specific ranking which is delegated to the user analysis. Another complementary thread of work investigates dynamic benchmarks where model training and data collection is interleaved to continually challenge model knowledge (56). To the best of our knowledge, this is the first work that studies the selection of multi-task, multi-domain benchmarks that satisfy limited compute budgets while maximizing research signal.

In summary, Table 1 shows the desiderata that we believe a good evaluation suite and framework should have such that they can both offer the community useful signal, and also balance that with being practical so that people can adopt it.

## 3 EEVEE METHODOLOGY

EEVEE is our proposed method for automating the selection of Pareto-optimal benchmark subsets. By analyzing benchmark performance metrics, EEVEE identifies a small, highly informative subset that maximizes information relative to the entire benchmark pool. This ensures that, as machine learning benchmark breadth and depth increases, we will always be able to identify and select few that offer high information about the whole. We strike a balance between providing rich evaluation signals and maintaining simplicity, reducing computational costs and human efforts required for adopting new benchmarks. EEVEE enables the production of a tiered evaluation engine accommodating various computational budgets, fostering an inclusive and accessible research environment, and improving the quality of insights derived from machine learning research while addressing reluctance towards resource-intensive evaluation processes. This balance between efficiency, simplicity, and signal richness presents EEVEE's value proposition for advancing machine learning research.

**Working Principle of EEVEE:** EEVEE works by building a *meta-model* over the performance metrics of models sufficient both in number and diversity, on the full benchmark pool from which we want to choose our subset. With the term *benchmark* in this paper we refer to a `dataset + task` pairs. The meta-model is parameterized as a 2-layer MLP network with 128 hidden neurons and leaky relu activation.

Formally, given a large benchmark pool $B = \{b_0, b_1, \ldots, b_K\}$, where $B$ is the full set of benchmarks, and $b_i$ are individual benchmarks therein, we have a sufficiently large and diverse pool of model performance metrics $M = \{m_0^0, m_1^0, \ldots, m_K^N\}$. Here, $m_i^j$ is the performance metric of model $j$ on benchmark $b_i$. We aim to discover a subset of $B$ of size $k$. This means $k$ total benchmarks make up the subset. If we build a meta-model $g(M_{selected}, \theta)$ to predict all of $M$ given only the selected subset $M_{selected}$, it should minimize the following loss:

$$L_{EEVEE} = MSE(M, g(M_{selected}, \theta)) \tag{1}$$

In this equation, MSE is the mean squared error. $M$ represents the full set of performance metrics of all our models on the full benchmark pool $B$. The term $g(M_{selected}, \theta)$ represents the predictions of the meta-model $g$ with parameters $\theta$ when it is given the performance metrics of all models from the selected subset of benchmarks $B_{selected}$, referred to as $M_{selected}$.

However, our main focus lies in the selected combination of performance metrics $M_{selected}$ that can generalize well on previously unseen models. To that end, we must split $M$ into train, validation and test sets, each consisting of performance metrics acquired from different models (e.g. train $\rightarrow$ ResNet50, ViT-Base, CLIP, and val $\rightarrow$ ResNext50, DINO, DeIT), and explicitly optimize the inner loop test loss rather than the training loss, while we use the validation loss to select the best meta-model for test. Hence the loss we wish to minimize is:

$$L_{EEVEE}^{test} = MSE(M^{test}, g(M_{selected}^{test}, \theta)) \tag{2}$$

We need a non-differentiable method for choosing the $k$ benchmarks in $M_{selected}$, since brute force becomes intractable very quickly, so we employ evolutionary methods to learn the $k$ selected benchmarks.

This results in a bi-level optimization, with an evolutionary method on the outer loop $e(B_{selected})$, where $e$ is the evolutionary method, and $B_{selected}$ are the benchmarks being selected – or indeed, the genes being optimized, and a small meta-model parameterized as a neural network $g(\theta)$ that receives a train/val split from $B_{selected}$ and trains itself to do the task described in Equation 1, after which process it is scored using the val set using the loss in Equation 2. Then, once a given candidate of benchmarks $B_{selected}$ is scored, in this way, the outer loop performs a tournament selection where only the top 50 candidates are preserved and mutated by removing one benchmark at random, and adding another at random. Each winning candidate mutates into 10 children, and the parent is also preserved in the gene pool, producing a gene pool with 550 candidates for every cycle. At initialization, we sample 1000 random combinations. We have found that 1000 is a good starting population that is both tractable to score and facilitates the necessary diversity that enables limited variation in results across several runs, showcasing convergent behaviour. We include full pseudocode showcasing all the details related to how we performed EEVEE for our experiments in Algorithm 1, 2 and 3 in Figure 1.

**Balancing the different metrics:** In a given set of tasks, domains and datasets there can be an imbalance in terms of how many metrics each one has and what types of metrics. Some metrics are higher-is-better, while others are lower-is-better. We follow a simple way to balance this out, which is, for a given metric that is higher-is-better we simply apply standard normalisation, and for those lower is better, we first reverse their polarity and apply standard normalization. Then, for a given dataset with meny metrics, we take the mean of those metrics, and, for a task containing many datasets, we take the mean of the mean of the per-dataset metrics. Therefore optimizing for what can be considered a per-task equally weighted reward. There are many other ways to do this, and those can depend on the context and what one is trying to achieve, but we chose this general one, since our context was such.

**Algorithm 1** Scoring

**Require:** Performance metrics $M$, Input metrics $M_{\text{selected}}$, Epochs $E = 20$, Hidden dimension $d_{\text{hidden}} = 100$, Learning rate $\alpha = 0.01$, Weight decay $\lambda = 0.01$, Optimizer type $\omega = $ "AdamW"
**Ensure:** Evaluation score mean(scores)
1: Convert data to tensors $x = M_{\text{selected}}$ and $y = M$
2: Normalize $x$ and $y$
3: Initialize ShuffleSplit cross-validation $kf$
4: Initialize empty list scores
5: **for** each train, val split in $kf$ **do**
6:    Divide $x$ into $x_{\text{train}}$ and $x_{\text{val}}$; $y$ into $y_{\text{train}}$ and $y_{\text{val}}$
7:    Build meta-model $g(\theta)$ with hidden dimension $d_{\text{hidden}}$
8:    Train $g(\theta)$ on $x_{\text{train}}$ and $y_{\text{train}}$ for $E$ epochs with learning rate $\alpha$, weight decay $\lambda$, and optimizer $\omega$
9:    Predict $y_{\text{pred}} = g(x_{\text{val}}, \theta)$
10:   Compute mean squared error score $= \text{MSE}(y_{\text{pred}}, y_{\text{val}})$
11:   Append score to scores
12: **end for**
13: **return** mean(scores)

**Algorithm 2** Mutation

**Require:** $B_{\text{selected}} \subset B$, $B = \{b_1, b_2, \ldots, b_K\}$
**Ensure:** New $B'_{\text{selected}}$
1: Select $b_{\text{remove}} \in B_{\text{selected}}$
2: Select $b_{\text{add}} \in B$
3: **while** $b_{\text{add}} \in B_{\text{selected}}$ **do**
4:    Select another $b_{\text{add}} \in B$
5: **end while**
6: Create $B'_{\text{selected}}$ by replacing $b_{\text{remove}}$ with $b_{\text{add}}$
7: **return** $B'_{\text{selected}}$

**Algorithm 3** Evolution

**Require:** Performance metrics $M = \{m_1^1, m_1^2, \ldots, m_K^N\}$, Benchmark set $B$, Combination size $k$, Number of winners $W$, Number of children per winner $C$, Number of generations $G$, Initial combinations size $I$, Training epochs $E$, Hidden dimension $d_{\text{hidden}} = 100$, Learning rate $\alpha = 0.01$, Weight decay $\lambda = 0.01$, Optimizer type $\omega = $ "AdamW"
**Ensure:** Evolved benchmark combinations $B_{\text{winners}}$
1: Initialize initial combinations $B_{\text{initial}}$ with $I$ random samples from $B$ of size $k$
2: Evaluate performance of $B_{\text{initial}}$ using SCOR-ING$(M, B_{\text{initial}}, E, d_{\text{hidden}}, \alpha, \lambda, \omega)$ and store scores in $S$
3: Select top $W$ combinations from $S$ as $B_{\text{winners}}$
4: **for** generation $g = 1$ to $G$ **do**
5:    Initialize a new set of combinations $B_{\text{new}}$
6:    **for** each combination $B_{\text{selected}} \in B_{\text{winners}}$ **do**
7:       Add $B_{\text{selected}}$ to $B_{\text{new}}$
8:       **for** each child $c = 1$ to $C$ **do**
9:          Mutate $B_{\text{selected}}$ using MUTATION$(B_{\text{selected}}, B)$ to create a new combination $B'_{\text{selected}}$
10:         Add $B'_{\text{selected}}$ to $B_{\text{new}}$
11:      **end for**
12:   **end for**
13:   Evaluate performance of $B_{\text{new}}$ using SCOR-ING$(M, B_{\text{new}}, E, d_{\text{hidden}}, \alpha, \lambda, \omega)$ and store scores in $S$
14:   Select top $W$ combinations from $S$ as $B_{\text{winners}}$
15: **end for**
16: **return** $B_{\text{winners}}$

Figure 1: (a) EEVEE Scoring algorithm, Mutation algorithm, and (b) Evolution algorithm.

**Architecture of Meta-Model:** We attempted deeper and shallower MLPs and transformers with various activations and hidden sizes but the chosen network balances speed of training with generalization. We have run experiments using 1-5 layer MLPs and transformers, with varying activation functions and hidden sizes ranging from 8-256. We found that a 2 layer MLP with 128 hidden size and leaky relu activation function offered the best generalization performance as a 2 layer transformer, but was much cheaper to train. Therefore, we used the 2-layer MLP throughout.

**Applying EEVEE on Model Encoder Generalization**

**Why Model Encoder Evaluation?** A common practice across machine learning applications involves augmenting general model encoders with task-oriented heads (13). The adaption of this paradigm can be attributed to the computational efficiency associated with training model encoders, over more expensive setups. Much of computer vision, as well as vision to text search and retrieval happen using model encoders (15; 77). Similarly, various applications requiring translation from one domain/modality/task to another require an encoder of some sort (34). Even the "decoder-only" LLM models that have demonstrated incredible capabilities in the last few years, internally can be seen as a series of representation encoders, a series of refinement before they reach the decoding stage (63).

Multi-modal early fusion is another topic closely related with model encoders – as research in early fusion can be done most efficiently when trying to learn data encoders rather than a full encoder-decoder, or decoder-only models (35). World model research in multi-modal dimensions can also take place most efficiently within a model-encoder context. Recent works like I/VJEPA (2) for example have paved the way for self-supervised learning which functions using model encoders, and has been demonstrated to be more efficient and more generalizable than full pixel decoding variants.

**The goal of focusing on Model Encoder Evaluation:** By applying EEVEE to search for a pareto-optimal set of benchmarks, *and* packaging it up in a unified framework that is built for the researcher in mind from the ground up, one which offers out of the box automated downloading, pipeline building, task adapters, and a very mature training and eval loop. Within this framework, we facilitate, all relevant logging information, including key training and eval metrics, rich gradient information, power and computational information, as well as visualizations where relevant. Finally, we support easy switching of model encoders, no matter what source modality they come from – our framework

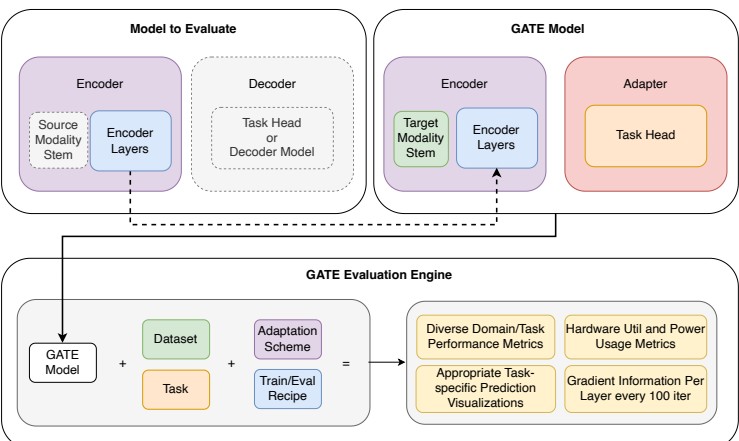

Figure 2: GATE Framework Pipeline

dubbed *GATE* is a one stop shop for ones model representation research needs, both during research, debugging, as well as at the evaluation phase.

GATE comes in three tiers *small*, *base* and *big*-GATE. Each having 8, 15 and 21 benchmarks within it, and targetted towards 12/24 and 36 GPU hours on a A100 40GB. We hope that by making it very easy for the end user and offering such rich signal for machine learning research, many researchers will choose to use GATE, to enhance their research signal, whilst keeping the compute budgets relatively feasible.

**Preparations: Choosing Models, Benchmarks and Adaptation Processes:** EEVEE will yield better results if the space of models, benchmarks and adaptation processes we use is diverse, but also thorough in numbers. **A. Adaptation Process** We wanted GATE to cover multiple domains, tasks and modalities when shifting from the source to the target setting. For that reason we decided that if a model encoder has an input layer that does not fit the target modality, we simply remove that input layer and replace it with a relevant ViT-like patchification (12) followed by a linear combination for each patch. For tasks where we have text, we would tokenize the text using BPE (54), and for tasks where we have video we would use the model encoder on each image, to acquire an image-level vector representation, and then follow that up with a simple 4 layer transformer that receives a sequence of image-vector tokens, to produce a video-level embedding, on top of which we apply the task-specific head at hand. The task-adapters we used leaned on established methods, and where possible we just used a transformer head, which includes segmentation, relational reasoning and video classification, with everything just using a linear head, full details available at H. After these modifications, described in Figure 2, we use a fine tuning scheme – this decision was informed by preliminary experiments on both full fine tuning and linear probe with a frozen backbone, in which we found that there was a clear superiority of fine tuning over linear probing for the benchmarks we chose in our pool. Full details of these preliminary experiments can be found in Appendix C.1. In our preliminary experiments we were able to identify three recipes, one for ConvNet-style architectures, one for ViT-style architectures and one for Hybrid architectures such as ConvNext and ResNext that worked well for all tasks, details in C.1.

**B. Model Pool** We wanted the space of models used to cover many important pretraining schemes, architectures, and source modalities. The details of these choices are provided next: **1. Pre-training Task and Dataset Variation**: With a consistent architecture, models were subjected to various pretraining tasks and datasets. Model instances representing this category include `CLIPViT` (46), `ConvNextV2` (38), `Siglip`, `FlexViT` (7), `LaionViT`, `ImageNet1K ViT` (11) with Random Augment, `SAM-ViT`, `DiNoViT`, `EfficientFormerV2` (33) and `DeiT3` (62). Further to these, we include models initialized from scratch, specifically, `ViT`, `ResNet50` (19), `FlexViT`, `EfficientNetV2` (60), and then fine-tuned on the GATE tasks. **2. Architectural Variation**: We explored models having the same pretraining dataset (ImageNet), but differing in their architecture. This group encompassed a mix of standard CNN models such as `EffNetV2`, `ResNet50`, `ResNext50` (71), `ConvNextV2_Base` (38) and transformer-based models like `EfficientFormer` (33) and `FlexViT` (7). **3. Modality and Dataset Variation**: This axis comprised models trained on modalities other than vision such as `Whisper`, coming from

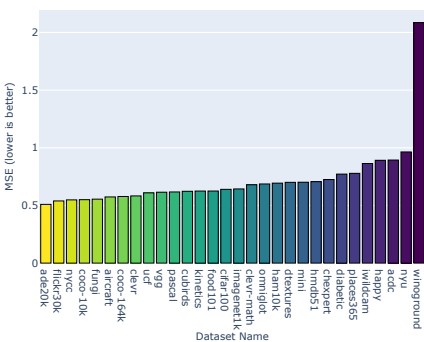

Figure 3: The EEVEE MSE Loss (k=1) shows "predictiveness over the whole," with lower values being better. Benchmarks like iWildcam, HappyWhale, and WinoGround test unique capabilities and may not predict all tasks, yet EEVEE often includes at least two of these in its top combinations along with a "natural-image representative" such as CIFAR100, ADE20K or Flickr30K.

an audio to text task and `Bert` (10), `Bart` (32) and `Mpnet` (58) coming from various text-based tasks. These models had their original input processing systems replaced by a Vision Transformer style embedding and were subsequently fine-tuned on the GATE tasks.

**C. Benchmark Pool** The benchmark pool, detailed in the Appendix, includes Image Classification (ImageNet1k (9), CIFAR100 (29), Places365 (79), Food101 (39), HappyWhale (18)), Few Shot Image Classification (Aircraft (40), Fungi (53), MiniImageNet (66), CUB200 (67), Describable Features (73)), Zero Shot Text-Image Classification (Flickr30K (44), New Yorker Caption Context (21), Winoground (61)), Visual Relational Reasoning (CLEVR (24), CLEVRMath (37)), Image Semantic Segmentation (ADE20K (80), COCO10K (36), COCO164K (36), NYU-Depth-v2 (57), PascalContext (41), Cityscapes (8)), Medical Image Classification (Chexpert (22), Diabetic Retinopathy (17), HAM10000 (64)), Medical Segmentation (ACDC (5)), Video Classification (HMDB51 (31), UCF-101 (59), Kinetics400 (25)) and Video Regression (iWildcam (4)).

**Producing Diverse Model Performance Metrics:** We apply our adaptation process on each and every model chosen, on every benchmark in the benchmark pool. To acquire test results we ensemble by averaging logits of the top 1, 3 and 5 validation models to produce three separate ensemble results.

**D. Experimental Approach** We wanted our research environment to reflect the end user, so we can properly understand their needs, and to offer a *pragmatic* experimental setup of in-the-wild researchers with little time to hyperparameter optimize, and which have to make decisions on small amounts of preliminary experiments – someone choosing a model encoder off the shelf and adapting it to downstream setting. For that reason, we kept any hyperparameter tuning, or human attention when it came to specific models to a minimum. Instead, we relied on existing good recipes, and did some preliminary experiments as explained in detail in C.1. Briefly, we discovered specific adjustments for each architecture type: for Convolutional Architectures, we used AdamW with a learning rate of 1e-3, and 6e-4 for segmentation tasks; for Vision Transformer Architectures, AdamW with a learning rate of 1e-5; and for Convolutional + Transformer Hybrid Architectures, AdamW with a learning rate of 2e-5. A plateau learning rate scheduler was configured with parameters like mode "min", factor 0.5, patience 1000, and threshold 1e-4, allowing models to effectively choose their own schedules on their learning progress. This adaptive scheduling facilitated "good enough" learning rates and enhanced performance across different architectures.

## 4 RESULTS

**Single Benchmark Predictiveness:** As demonstrated in Figure 3, using EEVEE we quantified the predictive power of each benchmark **on its own**, when not in a combination with others. We have found that ADE20K, Flickr30K, and the New York Caption Competition lead in their predictive power, with few-shot tasks, and relational reasoning, being very close to the best in predictive power. ImageNet1K sits squarely in the middle of the competition. Furthermore, some of the most "novel" benchmarks like iwildcam, happy whale, ACDC, NYU and Winoground are the least predictive tasks, Winoground being magnitudes less predictive. We argue that this is mainly due to the tasks being "harder", and our models being less designed for those. The results in WinoGround were bearly better than chance for example. However, when once we move to combinations of benchmarks, these 'less' predictive benchmarks become key contributors to better predictive power, as they represent edge cases, as can be seen in Figures 6g 7c, 7i in Appendix J, where these have the highest importance when removed from a given set.

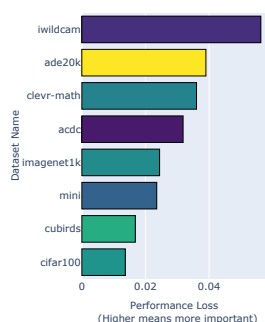

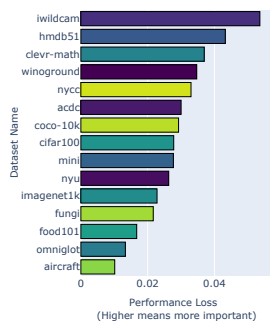

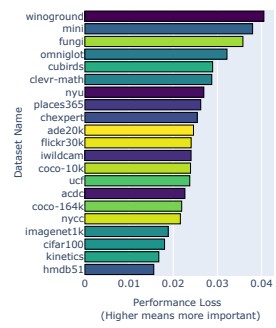

(a) smallGATE (k=8, 12 GPU hour) tier

(b) baseGATE (k=15, 24 GPU hour) tier

(c) bigGATE (k=21, 24 GPU hour) tier

Figure 4: Degradation of predictive power when a trained from scratch, for different GATE tiers.

**Predictiveness of Discovered Combinations** In Figure 5, we can see how the top-50 performing candidate combinations perform as we vary the number of benchmarks per combination from 1 to 26. We can see that there is a point of diminishing returns around the $k = 8$ point, after which there appears to be some "overfitting" occuring. We verified that the overfitting was a result of having a small sample number of 20 models, to train, val and test our meta-models with. We tried our level best to find the best architecture and regularization schemes for our meta-model, and this was the best we could do given available compute and (human) time. We

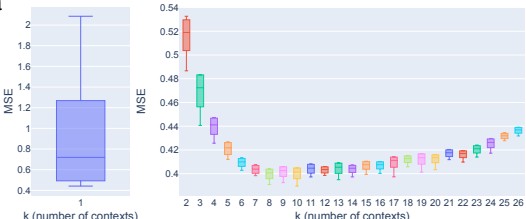

Figure 5: Performance of Models build with K-best datasets: We do a search over the space of all $k$ for EEVEE and box plot the population summary statistics of the top 50 combination candidates.

chose 8, 15, and 21 as the combination-threshold to make our packs out of as they satisfied the computational budgets we set for ourselves, and they have very diverse and predictive tasks, as can be seen in Figures 6g 7c, 7i. For full details on all the discovered top-k combinations please look at Appendix Section J.1. **Best Models based on GATE:** As can be seen in Table 2, or the Appendix extended Table 3, the best overall models are `ConvNextV2`, `SigLIP` and `CLIP` in that order, with SigLIP and CLIP often exchanging ranks between themselves. However, it is worth noting that `EfficientNetV2` demonstrated exceptional performance/compute across all tasks, and even outperformed all models in many medical tasks. Finally, ConvNet based models, and particularly ResNext50 seem to have done exceptionally well in the edge-case scenarios of ACDC, Happy Whale Individual identification, and general medical tasks, which indicates perhaps some sort of learning efficiency advantages related to their inductive biases.

**Limitations:** We empirically evaluatd EEVEE on a relatively large pool of models and benchmarks, however, with more models, and benchmarks it could yield much more general results. Especially with benchmarks targeting the text and audio modalities, as well as potentially offline RL.

## 5 CONCLUSION

In this paper, we propose EEVEE, an evolutionary-method-based search algorithm that can discover out of a large collection of benchmarks, the ones that can offer the most predictive value on the original collection, for a given set of models. We apply EEVEE on the task of model-encoder evaluation in the context of images, image-text, videos, and medical domains. As a result, we obtain the GATE Benchmark, which consists of 3 tiers, each targeted to a particular GPU budget, from 12, 24 and 36 GPU hours, per model evaluation. We then introduce the GATE engine, which takes these benchmarks, and offers a researcher-designed environment in which one can easily port their own model encoder, and run the full GATE tiers, and automatically produce a variety of performance, energy/power, hardware utilization metrics and task visualizations. We evaluated 20 representative models ranging from image, image-text, text and audio pretrained models, on the GATE tiers, and we discovered that ConvNextV2 and SigLIP seem to lead the pack overall, with EfficientNetV2 being an exceptional, efficient alternative for the medical domain and for *unique scenario* tasks, such as Happy Whale, ACDC and WinoGround. Finally, ConvNet based models, and ResNext50 in particular, seem

| Metric ↓ \| Model → | cvnxtv2 | siglip | clip | flex | deit | laion | vit | dino | smvit | rnx50 | effv2 | r50a1 | effrmr | seffv2 | sflex | svit | whspr | sr50a1 | bert | bart | mpnet |
|---|---|---|---|---|---|---|---|---|---|---|---|---|---|---|---|---|---|---|---|---|---|
| **Img Class** | | | | | | | | | | | | | | | | | | | | | |
| CIFAR-100 Acc@1 | 84.2 | 74.6 | 76.9 | 75.1 | 66.7 | 75.1 | 66.6 | 55.7 | 50.3 | 69.3 | 67.3 | 34.3 | 15.6 | 37.6 | 10.3 | 7.8 | 11.0 | 15.9 | 14.5 | 9.0 | 1.0 |
| Food-101 Acc@1 | 92.9 | 91.6 | 93.3 | 89.1 | 87.3 | 91.4 | 86.5 | 84.8 | 75.7 | 86.1 | 86.4 | 69.4 | 61.6 | 36.5 | 24.5 | 25.8 | 17.0 | 16.3 | 18.7 | 11.6 | 8.5 |
| HWhale Individual Acc@1 | 75.6 | 31.7 | 35.2 | 48.4 | 23.7 | 21.0 | 27.5 | 9.1 | 3.6 | 78.7 | 77.1 | 5.2 | 4.4 | 33.2 | 2.8 | 2.5 | 2.2 | 2.1 | 2.3 | 1.7 | 1.5 |
| HWhale Species Acc@1 | 99.8 | 99.8 | 99.7 | 99.8 | 99.5 | 99.7 | 99.7 | 99.2 | 95.4 | 99.7 | 99.7 | 92.1 | 92.8 | 96.5 | 76.5 | 74.5 | 64.3 | 65.8 | 71.2 | 59.3 | 62.9 |
| ImageNet-1K Acc@1 | 85.3 | 81.9 | 76.0 | 82.3 | 82.1 | 74.1 | 68.3 | 77.9 | 75.5 | 77.6 | 73.5 | 72.5 | 44.6 | 16.9 | 3.2 | 2.4 | 2.2 | 1.3 | 1.5 | 0.8 | 0.2 |
| ImageNet-1K Acc@5 | 96.8 | 95.8 | 93.7 | 95.5 | 94.7 | 93.1 | 89.1 | 93.0 | 90.8 | 93.3 | 91.4 | 90.5 | 72.5 | 37.3 | 10.1 | 8.2 | 7.7 | 4.7 | 5.2 | 3.2 | 1.2 |
| Places365 Acc@1 | 54.7 | 53.5 | 54.1 | 52.1 | 49.0 | 53.7 | 47.5 | 47.3 | 27.1 | 51.8 | 51.5 | 40.9 | 25.2 | 26.6 | 9.0 | 8.6 | 7.5 | 5.0 | 5.2 | 3.0 | 2.2 |
| **Task Mean** | 84.2 | 75.6 | 75.6 | 77.5 | 71.8 | 72.6 | 69.3 | 66.7 | 59.8 | 79.5 | 78.1 | 57.8 | 45.2 | 40.7 | 19.5 | 18.6 | 16.0 | 15.9 | 17.0 | 12.6 | 11.1 |
| **Few-Shot Img Class** | | | | | | | | | | | | | | | | | | | | | |
| Aircraft Acc@1 | 96.7 | 96.6 | 97.4 | 95.9 | 95.3 | 96.7 | 96.3 | 94.4 | 92.9 | 91.6 | 90.6 | 86.2 | 78.2 | 59.2 | 54.9 | 50.4 | 55.1 | 58.2 | 61.2 | 60.8 | 57.2 |
| CUBirds Acc@1 | 98.0 | 97.9 | 97.2 | 96.4 | 96.2 | 96.6 | 95.9 | 94.4 | 93.4 | 92.8 | 92.1 | 89.4 | 86.3 | 52.5 | 50.0 | 45.2 | 44.4 | 31.9 | 48.4 | 50.3 | 48.5 |
| DTextures Acc@1 | 85.0 | 85.2 | 88.6 | 78.9 | 81.9 | 86.1 | 80.8 | 79.4 | 81.9 | 77.7 | 60.3 | 77.2 | 68.5 | 46.6 | 50.2 | 50.5 | 50.0 | 33.1 | 44.6 | 49.8 | 38.3 |
| Fungi Acc@1 | 85.8 | 85.6 | 85.7 | 83.7 | 80.6 | 85.2 | 81.3 | 77.4 | 77.7 | 74.1 | 73.7 | 67.1 | 59.2 | 27.6 | 38.0 | 37.0 | 33.9 | 28.2 | 32.9 | 33.8 | 7.6 |
| Mini-Imagenet Acc@1 | 97.0 | 96.2 | 93.1 | 99.1 | 98.8 | 90.8 | 89.9 | 98.7 | 92.9 | 94.1 | 63.2 | 93.2 | 90.9 | 36.7 | 45.9 | 47.2 | 44.8 | 34.2 | 39.7 | 37.3 | 36.8 |
| Omniglot Acc@1 | 98.6 | 98.9 | 99.0 | 98.9 | 98.7 | 98.9 | 98.8 | 98.6 | 98.6 | 98.5 | 98.7 | 95.5 | 95.8 | 98.2 | 93.4 | 93.6 | 82.9 | 80.5 | 90.2 | 84.1 | 90.7 |
| VGG Flowers Acc@1 | 99.7 | 98.9 | 98.6 | 96.7 | 96.2 | 97.0 | 95.9 | 95.5 | 93.4 | 87.9 | 91.3 | 89.3 | 90.6 | 59.6 | 69.4 | 69.4 | 63.0 | 53.4 | 59.1 | 59.4 | 60.8 |
| **Task Mean** | 94.4 | 94.2 | 94.2 | 92.8 | 92.5 | 93.1 | 91.3 | 91.2 | 90.1 | 88.1 | 81.4 | 85.4 | 81.4 | 54.3 | 57.4 | 56.2 | 53.4 | 45.6 | 53.7 | 53.6 | 48.6 |
| **Img Seg** | | | | | | | | | | | | | | | | | | | | | |
| ADE20K mIoU | 46.8 | 47.1 | 44.0 | 43.7 | 37.8 | 43.4 | 33.2 | 33.3 | 25.9 | 18.2 | 14.2 | 11.7 | 9.8 | 1.5 | 0.5 | 0.4 | 0.6 | 0.4 | 0.4 | 0.5 | 0.4 |
| Cityscapes mIoU | 62.3 | 69.8 | 67.6 | 67.5 | 63.9 | 67.7 | 63.9 | 61.4 | 59.5 | 40.8 | 64.2 | 40.2 | 2.5 | 46.7 | 22.8 | 23.5 | 17.1 | 18.6 | 2.7 | 2.0 | 2.7 |
| COCO-10K mIoU | 26.9 | 39.5 | 35.6 | 35.1 | 32.8 | 33.6 | 29.8 | 31.0 | 28.6 | 18.4 | 10.2 | 5.7 | 14.0 | 1.1 | 0.9 | 0.8 | 0.4 | 1.6 | 0.1 | 1.3 | 0.1 |
| COCO-164K mIoU | 32.7 | 36.7 | 33.8 | 33.0 | 30.5 | 32.4 | 27.0 | 28.9 | 25.7 | 16.8 | 9.7 | 4.7 | 13.7 | 1.0 | 0.7 | 0.7 | 0.5 | 0.7 | 0.1 | 1.1 | 0.1 |
| NYU mIoU | 7.5 | 7.7 | 7.8 | 6.9 | 12.2 | 5.7 | 6.1 | 12.1 | 11.0 | 5.9 | 8.3 | 6.4 | 10.5 | 6.8 | 3.5 | 3.7 | 2.9 | 7.2 | 5.4 | 5.0 | 5.4 |
| Pascal mIoU | 32.8 | 34.8 | 35.7 | 30.6 | 31.4 | 28.3 | 27.5 | 29.8 | 24.0 | 16.6 | 11.7 | 6.8 | 14.0 | 1.7 | 1.3 | 1.1 | 1.4 | 2.3 | 1.0 | 1.4 | 0.9 |
| **Task Mean** | 34.8 | 39.3 | 37.4 | 36.2 | 34.8 | 35.2 | 31.3 | 32.8 | 29.1 | 19.5 | 19.7 | 12.6 | 10.8 | 9.8 | 4.9 | 5.0 | 3.8 | 5.1 | 1.6 | 1.9 | 1.6 |
| **Img Relational** | | | | | | | | | | | | | | | | | | | | | |
| CLEVR Acc@1 | 52.5 | 52.7 | 52.7 | 52.1 | 52.6 | 52.6 | 52.8 | 52.8 | 51.6 | 50.1 | 40.6 | 49.3 | 45.2 | 39.3 | 46.1 | 45.9 | 46.4 | 44.9 | 42.6 | 42.5 | 41.2 |
| CLEVR Colour | 35.4 | 36.1 | 36.4 | 35.0 | 35.5 | 35.6 | 35.3 | 36.1 | 34.2 | 26.8 | 15.7 | 24.7 | 14.7 | 12.5 | 25.7 | 29.4 | 28.8 | 22.8 | 13.2 | 13.0 | 13.2 |
| CLEVR Count | 45.8 | 45.8 | 45.8 | 45.9 | 45.8 | 45.7 | 45.7 | 45.6 | 45.6 | 45.3 | 39.0 | 45.1 | 44.8 | 37.9 | 45.1 | 44.7 | 44.8 | 44.9 | 44.7 | 44.7 | 43.0 |
| CLEVR Material | 60.5 | 60.6 | 60.5 | 60.0 | 60.5 | 60.6 | 61.4 | 61.3 | 60.2 | 58.6 | 52.1 | 57.5 | 53.7 | 49.8 | 53.7 | 51.7 | 54.0 | 53.0 | 49.8 | 50.5 | 49.9 |
| CLEVR Shape | 52.1 | 52.4 | 52.5 | 51.1 | 52.2 | 52.4 | 52.9 | 51.2 | 49.9 | 50.2 | 34.3 | 50.2 | 44.8 | 33.3 | 35.8 | 34.9 | 36.1 | 34.6 | 34.6 | 33.7 | 33.4 |
| CLEVR Size | 61.0 | 61.1 | 61.3 | 60.7 | 61.1 | 60.8 | 62.0 | 62.3 | 60.9 | 59.6 | 53.5 | 58.3 | 55.7 | 50.6 | 56.2 | 55.2 | 55.2 | 54.6 | 54.2 | 54.1 | 50.1 |
| CLEVR Yes/No | 60.7 | 60.5 | 60.8 | 60.6 | 60.5 | 60.7 | 60.4 | 60.4 | 60.2 | 59.8 | 53.3 | 59.9 | 59.6 | 51.4 | 60.1 | 59.2 | 59.5 | 59.8 | 59.5 | 59.3 | 58.6 |
| CLEVR-Math Acc@1 | 79.3 | 65.9 | 68.8 | 59.9 | 73.7 | 62.9 | 60.5 | 59.3 | 58.3 | 55.6 | 44.0 | 56.0 | 56.6 | 30.2 | 46.9 | 46.5 | 46.2 | 45.7 | 44.8 | 42.1 | 36.4 |
| **Task Mean** | 55.9 | 54.4 | 54.9 | 53.1 | 55.2 | 53.9 | 53.9 | 53.6 | 52.6 | 50.8 | 41.6 | 50.1 | 46.9 | 38.1 | 46.2 | 45.9 | 46.4 | 45.0 | 42.9 | 42.5 | 40.7 |
| **Medical Class** | | | | | | | | | | | | | | | | | | | | | |
| Chexpert APS Macro | 61.6 | 61.0 | 61.2 | 62.6 | 62.3 | 60.9 | 61.2 | 59.9 | 61.5 | 59.8 | 60.2 | 54.1 | 55.2 | 48.0 | 33.9 | 34.1 | 34.3 | 35.7 | 36.9 | 33.7 | 33.0 |
| Chexpert AUC Macro | 82.5 | 82.5 | 82.3 | 83.2 | 82.9 | 82.5 | 82.4 | 81.8 | 82.8 | 81.1 | 81.9 | 79.1 | 79.9 | 74.7 | 64.7 | 65.1 | 65.5 | 67.0 | 67.6 | 65.3 | 64.9 |
| Chexpert BS Macro | 84.3 | 84.4 | 84.5 | 85.1 | 86.2 | 84.6 | 84.9 | 85.6 | 87.0 | 86.3 | 84.8 | 86.1 | 86.4 | 84.6 | 82.9 | 82.9 | 83.0 | 83.1 | 83.1 | 82.8 | 82.8 |
| Diabetic APS Macro | 56.9 | 57.2 | 56.4 | 56.3 | 54.2 | 56.4 | 54.4 | 51.9 | 45.2 | 55.6 | 58.7 | 35.5 | 36.6 | 20.6 | 21.6 | 21.5 | 22.5 | 23.3 | 22.4 | 21.2 | 21.3 |
| Diabetic AUC Macro | 87.5 | 86.7 | 86.0 | 85.7 | 85.0 | 85.3 | 84.7 | 83.8 | 81.2 | 85.6 | 86.1 | 76.0 | 79.0 | 53.4 | 55.7 | 55.7 | 57.8 | 61.3 | 59.4 | 55.1 | 54.0 |
| Diabetic BS Macro | 94.5 | 94.0 | 93.9 | 93.9 | 93.8 | 93.6 | 93.7 | 93.6 | 93.0 | 93.9 | 94.2 | 92.3 | 92.6 | 91.6 | 91.3 | 91.4 | 91.4 | 91.5 | 91.8 | 91.6 | 91.6 |
| HAM10K APS Macro | 94.5 | 93.3 | 91.4 | 92.2 | 91.3 | 92.1 | 91.6 | 90.8 | 83.4 | 87.9 | 87.1 | 43.7 | 46.9 | 38.8 | 38.0 | 35.9 | 32.2 | 48.5 | 50.6 | 37.6 | 32.6 |
| HAM10K AUC Macro | 99.1 | 98.6 | 98.7 | 98.5 | 98.6 | 98.6 | 98.7 | 98.5 | 97.8 | 97.9 | 97.5 | 89.3 | 90.1 | 85.6 | 86.1 | 84.6 | 82.8 | 91.0 | 91.1 | 85.9 | 83.3 |
| HAM10K BS Macro | 98.4 | 98.1 | 97.8 | 98.1 | 98.0 | 97.9 | 97.9 | 97.9 | 97.2 | 97.6 | 97.2 | 95.2 | 95.5 | 94.6 | 94.5 | 94.4 | 94.3 | 95.0 | 95.2 | 94.4 | 94.2 |
| **Task Mean** | 84.4 | 84.0 | 83.6 | 83.9 | 83.6 | 83.6 | 83.3 | 82.6 | 81.0 | 82.9 | 83.1 | 72.4 | 73.6 | 65.8 | 63.2 | 62.9 | 62.6 | 66.3 | 66.4 | 63.1 | 62.0 |
| **Medical Seg** | | | | | | | | | | | | | | | | | | | | | |
| ACDC Dice Score | 63.1 | 48.1 | 51.3 | 45.9 | 43.8 | 48.0 | 50.4 | 47.7 | 44.6 | 44.2 | 61.0 | 40.2 | 18.7 | 46.0 | 16.5 | 18.5 | 32.2 | 28.7 | 23.2 | 26.2 | 25.3 |
| **Task Mean** | 63.1 | 48.1 | 51.3 | 45.9 | 43.8 | 48.0 | 50.4 | 47.7 | 44.6 | 44.2 | 61.0 | 40.2 | 18.7 | 46.0 | 16.5 | 18.5 | 32.2 | 28.7 | 23.2 | 26.2 | 25.3 |
| **Img to Txt ZS** | | | | | | | | | | | | | | | | | | | | | |
| Flickr30K Img2Txt | 6.3 | 6.3 | 7.0 | 5.9 | 5.6 | 6.8 | 5.9 | 5.2 | 4.5 | 4.1 | 3.7 | 4.7 | 4.2 | 1.6 | 1.8 | 2.0 | 1.9 | 2.0 | 1.9 | 1.8 | 1.6 |
| Flickr30K Txt2Img | 5.7 | 5.9 | 6.0 | 5.3 | 5.1 | 6.5 | 6.0 | 5.1 | 5.0 | 3.8 | 4.0 | 4.2 | 3.9 | 1.7 | 1.8 | 2.0 | 2.2 | 2.3 | 1.9 | 1.7 | 1.6 |
| NYCC Img2Txt | 6.9 | 6.6 | 6.9 | 5.8 | 6.5 | 6.9 | 6.4 | 6.0 | 4.7 | 4.9 | 4.1 | 4.6 | 4.2 | 1.6 | 2.1 | 1.8 | 1.9 | 2.1 | 2.0 | 1.6 | 1.6 |
| NYCC Txt2Img | 6.1 | 5.9 | 6.4 | 5.5 | 6.0 | 6.2 | 6.4 | 5.8 | 4.8 | 4.3 | 4.1 | 3.9 | 3.7 | 1.6 | 2.0 | 1.7 | 2.0 | 2.4 | 1.9 | 1.8 | 1.6 |
| Winoground Img2Txt | 51.0 | 53.4 | 59.5 | 49.7 | 50.0 | 50.3 | 49.5 | 43.5 | 53.8 | 61.9 | 50.0 | 48.9 | 47.3 | 43.9 | 50.0 | 41.3 | 50.0 | 53.2 | 49.6 | 50.1 | 50.4 |
| Winoground Txt2Img | 50.0 | 55.2 | 56.2 | 53.1 | 50.0 | 55.5 | 48.3 | 54.2 | 48.6 | 54.8 | 50.0 | 49.6 | 52.4 | 52.8 | 50.0 | 54.2 | 51.8 | 52.2 | 51.8 | 48.8 | 52.1 |
| **Task Mean** | 21.0 | 22.2 | 23.7 | 20.9 | 20.5 | 22.0 | 20.4 | 20.0 | 20.2 | 22.3 | 19.3 | 19.3 | 19.3 | 17.2 | 18.0 | 17.2 | 18.3 | 19.0 | 18.2 | 17.6 | 18.1 |
| **Video Class** | | | | | | | | | | | | | | | | | | | | | |
| HMDB-51 Acc@1 | 52.5 | 40.7 | 40.6 | 32.2 | 39.3 | 24.9 | 27.4 | 32.8 | 33.1 | 5.6 | 11.5 | 1.8 | 2.1 | 3.8 | 8.3 | 7.9 | 6.1 | 5.4 | 6.4 | 7.5 | 4.0 |
| Kinetics Acc@1 | 48.8 | 44.2 | 51.4 | 43.7 | 40.3 | 44.6 | 33.2 | 36.4 | 25.8 | 2.7 | 1.0 | 0.2 | 0.3 | 0.4 | 2.0 | 1.6 | 1.0 | 0.5 | 0.3 | 0.3 | 0.3 |
| UCF-101 Acc@1 | 84.4 | 75.1 | 69.9 | 63.2 | 75.0 | 63.4 | 58.8 | 66.6 | 48.7 | 19.7 | 11.1 | 2.8 | 0.8 | 2.1 | 15.2 | 13.3 | 6.6 | 8.7 | 6.5 | 7.0 | 2.7 |
| **Task Mean** | 61.9 | 53.3 | 54.0 | 46.4 | 51.5 | 44.3 | 39.8 | 45.2 | 35.9 | 9.4 | 7.8 | 1.6 | 1.1 | 2.1 | 8.5 | 7.6 | 4.6 | 4.9 | 4.4 | 4.9 | 2.3 |
| **Video Reg** | | | | | | | | | | | | | | | | | | | | | |
| IWildCam MAE Score | 55.2 | 53.1 | 56.0 | 54.9 | 54.1 | 46.1 | 52.1 | 49.1 | 45.3 | 34.6 | 35.8 | 37.3 | 13.9 | 29.6 | 41.3 | 39.3 | 36.3 | 40.3 | 27.5 | 38.7 | 29.2 |
| **Task Mean** | 55.2 | 53.1 | 56.0 | 54.9 | 54.1 | 46.1 | 52.1 | 49.1 | 45.3 | 34.6 | 35.8 | 37.3 | 13.9 | 29.6 | 41.3 | 39.3 | 36.3 | 40.3 | 27.5 | 38.7 | 29.2 |
| **GATE** | | | | | | | | | | | | | | | | | | | | | |
| Full GATE Mean | 69.0 | 66.8 | 66.8 | 64.6 | 64.3 | 63.4 | 62.1 | 62.2 | 58.5 | 56.3 | 54.4 | 48.4 | 42.8 | 39.6 | 37.5 | 37.2 | 36.2 | 36.9 | 35.0 | 34.9 | 31.8 |
| Big GATE Mean | 76.6 | 74.5 | 74.4 | 72.8 | 72.0 | 71.9 | 70.6 | 70.0 | 66.8 | 66.7 | 64.8 | 58.5 | 53.1 | 46.8 | 43.8 | 43.4 | 41.9 | 41.5 | 40.9 | 39.8 | 37.1 |
| Base GATE Mean | 68.3 | 65.6 | 65.7 | 62.6 | 63.7 | 60.7 | 60.2 | 60.7 | 58.6 | 55.1 | 53.5 | 48.2 | 42.8 | 38.0 | 36.5 | 36.3 | 35.4 | 36.6 | 34.8 | 34.8 | 30.4 |
| Small GATE Mean | 77.7 | 74.9 | 74.6 | 73.3 | 72.4 | 71.2 | 68.9 | 69.1 | 65.3 | 65.7 | 61.7 | 58.5 | 49.3 | 40.5 | 35.7 | 35.4 | 35.9 | 35.3 | 34.1 | 34.4 | 30.4 |
| Full GATE Rank | 1.0 | 3.0 | 2.0 | 4.0 | 5.0 | 6.0 | 8.0 | 7.0 | 9.0 | 10.0 | 11.0 | 12.0 | 13.0 | 14.0 | 15.0 | 16.0 | 18.0 | 17.0 | 19.0 | 20.0 | 21.0 |
| Big GATE Rank | 1.0 | 2.0 | 3.0 | 4.0 | 5.0 | 6.0 | 7.0 | 8.0 | 9.0 | 10.0 | 11.0 | 12.0 | 13.0 | 14.0 | 15.0 | 16.0 | 17.0 | 18.0 | 19.0 | 20.0 | 21.0 |
| Base GATE Rank | 1.0 | 3.0 | 2.0 | 5.0 | 4.0 | 7.0 | 8.0 | 6.0 | 9.0 | 10.0 | 11.0 | 12.0 | 13.0 | 14.0 | 16.0 | 17.0 | 18.0 | 15.0 | 20.0 | 19.0 | 21.0 |
| Small GATE Rank | 1.0 | 2.0 | 3.0 | 4.0 | 5.0 | 6.0 | 8.0 | 7.0 | 10.0 | 9.0 | 11.0 | 12.0 | 13.0 | 14.0 | 16.0 | 17.0 | 15.0 | 18.0 | 20.0 | 19.0 | 21.0 |

Table 2: Summary of experiments: Black/Bold best model, Green second best, Blue third best, and red the worst performing model. Models prefixed with 's' refer to 'from scratch' trained models, rather than pretrained. For the full table look at Appendix Table 3

to have a lot more *learning* efficiency, as they are the best adapted models on very novel domains, such as Happy Whale individual prediction challenge, ACDC and medical tasks.

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

## A  END-USER GUIDELINES

For an end-user to use GATE, they need to:

1. Install the GATE framework python package, as described in the Github repo's readme page.

2. Choose a path for implementing the new foundation model encoder they wish to evaluate. This is either cloning the full GATE repo and modifying existing components directly, or, importing the `GATEncoder` and `GATEModel` classes from GATE, and wrapping up their model within it. Doing so requires the researcher to implement a relevant forward function that can take in the modalities their model needs to process, as well as defining a configuration that tells GATE what modalities a model can receive and output features on, as well as any transforms needed for a batch to be ready for their model.

3. The user chooses a GATE tier to use (from `smallGATE`, `baseGATE` and `bigGATE`). Based on the configuration defined by the user in step 2.

4. GATE generates a list of commands, each representing an experiment that needs to be run, and can then run these commands on your local GPU box, parallelizing the tasks, one on each available GPU, or, can provide a list of commands or json file that one can use to run these commands on a GPU cluster, or other hardware.

5. GATE emits a wandb project, with metrics, visualizations and other measures, allowing easy tracking of experiments, and sharing thereof, as well as huggingface model weights for each model being trained – which is also used to achieve a *stateless* execution.

6. Once the experiments are completed, one can invoke the `produce-analysis.py` file within GATE to get tables and figures that analyse the data, similar to what appears in this paper. Those results can then be used to report results in a paper, or, be used to make decisions for production models.

This process ensures the GATE framework is aware of what a model's supported modalities are, as well as how to produce modality-specific features, given the model. Once this is completed, the user, with a single line of code, can select a GATE tier, and launch all jobs needed to produce results for that tier. Importantly, GATE is made to facilitate and encourage foundation models that are diverse in their capabilities, and allow the researchers to focus on what matters – that is, designing and training their foundation model – rather than spending the majority of their time building and optimizing evaluation boilerplate. Furthermore, the diversity of signal that GATE provides allows better understanding of a given model's strengths and weaknesses, which as a result makes the research, review and iteration process of the field as a whole more efficient. This is because there is a consistent boilerplate that runs all models, with broad signal that reduces probability of making erroneous conclusions – both in the overly optimistic, or overly pessimistic side of things.

### A.1  PRINCIPAL USE CASES

1. **Model Development and Iteration**: GATE serves as a valuable tool during the model research and development phase. By integrating the model into GATE and running either the `smallGATE` or `baseGATE` tiers, developers can obtain a comprehensive and robust performance evaluation of their model across diverse domains, tasks, and modalities. Worth noting that GATE allows easy inclusion of foundation models **pretrained on images, video, audio, text, etc**, to be **fine-tuned on pixel-based tasks**. It achieves this by replacing a model's root layer / embedding layer, with one appropriate for a given task's modality, and adding on top a relevant task adapter head.

2. **Model Evaluation for Machine Learning Research**: GATE enhances the communication of research findings and their potential applications, a vital aspect of scientific collaboration. By using GATE as a benchmark, even at the most cost-efficient GPU hour level of `smallGATE`, the clarity and depth of future ML papers can be significantly improved. GATE's explicit evaluation of modality, domain, and task shifts in a given foundation model provides a nuanced and informative perspective on a model's true capabilities, offering a more detailed understanding of a model's strengths and weaknesses than optimizing a single metric, such as ImageNet validation error.

## B  RESULT EXTRAS

The results were logged in `WandB`, and then further processed after all experiments were completed to generate the tables and figures in this paper. Much of the logged information outside of testing metrics were not used for any of the figures and tables in this paper. The full set of experiments and all the logged results can be found at our wandb gate project repo[2].

### B.1  RESULT PROCESSING

Once all experiments were completed, we queried our wandb project repository and returned test results from all our experiments, if an experiment name was duplicated, we used the latest entries, and, for each experiment type there existed three independent runs. We averaged the results of any metrics across such independent runs to acquire a better approximation to the true performance of those models.

## C  PRELIMINARY EXPERIMENTS DETAILS

### C.1  PRELIMINARY EXPERIMENTS

First, we trained models on ImageNet1k, CIFAR100, CLEVR, ADE20K, CityScapes, and, ACDC for 5K iterations, using cosine annealing learning schedule or plateau annealing, with AdamW, weight decays varying from 0.1 - 0.0001, and applied models from each major architecture category – specifically, the CLIPViT, ImageNet pretrained ViT, ResNext, ResNet and ConvNextV2. The results from these experiments pointed to the fact that there exists one general and good recipe for each architecture style. The recipes that we discovered were as follows:

#### C.1.1  ACROSS ARCHITECTURE SETTINGS

Unless otherwise stated, the settings here are applied universally in all experiments.

**Optimizer**: AdamW, weight decay 0.01, plateau annealing with patience 1000, relative scaling and scale factor 0.5, and, threshold 0.0001.

**Training Details**: Training iterations: 10K, validate every 500 iterations.

**Test Details**: Top-3 validation models (across all validated checkpoints) are ensembled by prediction averaging.

#### C.1.2  ARCHITECTURE SPECIFIC SETTINGS

**Convolutional Architectures**: **Optimizer**: AdamW, learning rate 1e-3, and for segmentation tasks only, we used learning rate 6e-4

**Vision Transformer Architectures**: **Optimizer**: AdamW, learning rate 1e-5

**Convolutional + Transformer Hybrid Architectures Optimizer**: AdamW, learning rate 2e-5

The above recipes were what we used throughout all our experiments unless otherwise stated.

## D  GATE GUIDING PRINCIPLES

The fundamental values driving the design decisions behind GATE are the following:

1. Maximizing Generalization Signal: GATE is designed to provide a high signal-to-noise ratio concerning a model's ability to generalize in diverse downstream contexts, that vary in domain, task and modality. This allows for a more robust assessment of a model's capacity for adaptation and versatility. By noise here we refer to how clear a given signal response is. For example, an image classification test accuracy signal on ImageNet, would provide clear

---

[2]omitted until double blind is over

signal with respect to the natural domain and the classification task, but would be blurry for more compositional, object disentanglement and relational tasks, such as segmentation, or, visual question answering.

2. Time Efficiency: Acknowledging the importance of computational resources and time, GATE operates within set benchmarks of 12, 24, and 36 GPU hours (established on A100 @ 40GB). These set timeframes ensure GATE's assessments are both thorough and expedient.

3. Minimizing Usage Friction: The framework supporting GATE is designed to be user-friendly, enabling easy integration of new backbones and facilitating smooth experimentation. This low-friction approach ensures a streamlined experience when using GATE, making the process of evaluation more efficient.

We argue that a good balance of the above can generate a pragmatic, yet thorough foundation model evaluation suite, that will, importantly, be of real use to most researchers in the field.

## E  DEFINING THE GATE BENCHMARK

GATE is a comprehensive evaluation engine designed to advance the development of more general machine learning models. It improves on existing benchmarks by enabling the evaluation of models across diverse modalities, domains, and tasks.

GATE is composed of three key components. The first is a benchmark *pool*, a broad collection of datasets, tasks, and processes that measure a model's performance across various domains, tasks, and modalities. The second component is a set of benchmark *tiers*, which are meticulously curated subsets from the GATE benchmark pool, tailored to specific compute budgets and project phases. The final, and is a software framework, designed to seamlessly integrate new foundation models and execute the GATE tiers, thereby enabling efficient performance evaluation across a diverse range of downstream modalities, domains, and tasks. Practically, GATE is directed towards machine learning researchers and developers as a means to efficiently, and with little friction, get broad signal about how their model performs after transfer in diverse contexts, specifically selected for their empirically evaluated high signal-to-noise ratio with respect to predictive power in how a model performs in previously unseen contexts.

Building GATE was a careful balancing act. We needed to respect specific time budgets while also aiming for a wide variety of evaluation scenarios. Our approach was as follows:

1. Select a diverse set of learning contexts, spanning multiple domains, tasks and modalities. We refer this as the *Benchmark Pool*.

2. Select a broad set of key foundation models, varying in their architecture, pretraining scheme and source modality. We refer to this as the *Model Pool*.

3. Fine tune each of the models in the model pool, on each of the contexts in the benchmark pool. Evaluate trained models on each context's test sets.

4. Use the test set results acquired to quantify the predictive power each benchmark holds with respect to previously unseen benchmarks, both at the individual level and the collection level. We call this measure, the *downstream generalization predictability measure* (**DGPM**).

5. Use the DGPM values of the various combinations of benchmarks to build the three GATE tiers, selecting combinations of benchmarks that can provide the most information within a target time budget.

We elaborate on each of the above steps in the following subsections.

## F  BENCHMARK POOL SELECTION DETAILS

**Medical Image Classification**: Medical data are known to present a substantial shift in both domain and even modality depending on their format. We have selected datasets that not only pose significant challenges for foundation models but also align with the broader imperative to deliver real-world benefits downstream.

***Chexpert***: A dataset comprising a challenging array of chest x-rays annotated with findings critical to diagnosing thoracic diseases. It tests models on their ability to navigate complex, multi-label medical data, encapsulating the kind of nuanced decision-making that AI must augment in clinical settings.

***Diabetic Retinopathy Classification***: Early detection of diabetic retinopathy from retinal images is a public health priority; models fine-tuned on this dataset can have immediate implications for preventing vision loss on a global scale. This dataset requires models to decipher fine-grained, progressive changes indicative of the disease, reflecting the precision necessary for medical AI applications.

***HAM10000 (Human Against Machine with 10000 dermatoscopic images)***: The dataset provides a diverse spectrum of skin lesion images vital for differentiating between benign and malignant conditions. Incorporating this dataset not only challenges the pattern recognition prowess of AI but also contributes to the advancement of dermatology through machine learning technologies.

**Metrics**: We collect Average Precision Score (APS), Area Under the Receiver Operating Characteristics Curve (AUC), and Brier Score (BS) both overall (i.e. macro) as well as for individual pathologies/classes.

**Medical Segmentation**: This category evaluates foundational models' ability to generalize from natural to medical image modalities and to perform domain-specific tasks that require precision and complex spatial understanding:

***ACDC (Automated Cardiac Diagnosis Challenge)***: This dataset is aimed at assessing models' generalization to the medical domain, particularly the transferability of representations for segmenting anatomical structures in cardiac MRI images. By focusing on the heart's intricate anatomy, ACDC tests the models' ability to adapt to clinically relevant shapes and patterns—a shift from common visual recognition tasks to precise medical delineation. **Metrics:** We collect dice loss, mIoU, mean accuracy and overall accuracy.

## G   BENCHMARK POOL DETAILS

Having a set of diverse benchmarks ranging in challenge factor, as well as modality, task and domain shift was key. We explain in more detail why why consider these factors important in Appendix in more detail. We refer to this as our *benchmark pool*, and it consists of the following:

**Image Classification:** We employ **ImageNet1k** (9), **CIFAR100** (29), **Places365** (79), and **Food101** (39) to cover diverse natural image domains. Additionally, we include **HappyWhale** (18) for a more challenging domain shift, aiding in wildlife research and providing an interesting test case for model evaluation.

**Few Shot Image Classification:** We use the MetaDataset task recipe on the **Aircraft** (40), **Fungi** (53), **MiniImageNet** (66), **CUB200** (67), and **Describable Features** (73) datasets to evaluate task and domain shift robustness for an evaluation model.

**Zero Shot Text-Image Classification:** Another key setting is that of zero-shot text-image classification, on which many current key models were trained and evaluated (46). We utilize **Flickr30K**, **New Yorker Caption Context** (a challenging humor task), and **Winoground**–a task requiring the model to match two texts with their corresponding images, focusing on compositional differences.

**Visual Relational Reasoning**: A context where earlier models, such as ResNet50 (19) had low performance without layers with associative inductive biases (e.g., relational neural networks or transformers (52; 65)). This ensures we are aware of any trade-offs in relational compositional abilities in our models. We use **CLEVR** (24) and **CLEVRMath** (37).

**Image Semantic Segmentation**: Essential for various real-world applications, serving as an indicator of a model's ability to retain spatial information and identify objects at a per-pixel level. **ADE20K** (80), **COCO10K** (36), **COCO164K** (36), **NYU-Depth-v2** (57), **PascalContext** (41), and **Cityscapes** (8).

**Medical Image Classification**: Medical data exhibit substantial domain and modality shifts, posing significant challenges for machine learning models while aligning with the imperative to deliver real-world benefits.***Chexpert*** (22) (chest X-rays annotated for thoracic disease diagnosis), ***Dia-***

*betic Retinopathy Classification* (17) (retinal images for early detection of diabetic retinopathy), *HAM10000* (64) (dermatoscopic images for differentiating skin lesions).

**Medical Segmentation** → *ACDC (Automated Cardiac Diagnosis Challenge)* (5): This dataset assesses models' generalization to the medical domain, particularly the transferability of representations for segmenting anatomical structures in cardiac MRI images. By focusing on the heart's intricate anatomy, ACDC tests the models' ability to adapt to clinically relevant shapes and patterns.

**Video Classification**: Video classification tasks test models on their temporal generalization abilities and require an understanding of not only individual frame content but also the transition and context between frames. *HMDB51 (Human Motion Database)* (31), *UCF-101 (University of Central Florida - 101 action categories)* (59), *Kinetics400* (25).

**Video Regression**: Where classification tasks gauge categorical distinctions, video regression tasks assess models' ability to make continuous numerical predictions from temporal data, serving as an indicator of a model's capability to process and quantify dynamic content. *iWildcam (International Wildlife Camera Trap Challenge)* (4): This dataset targets estimating animal species abundance from videos and is a direct test of modality and task shift, and showcases a models' potential impact on ecological monitoring and species conservation efforts.

1. **Modality shifting** contexts: Contexts where the foundation model is asked to learn to do well at a task that requires understanding of a previously unseen modality. More specifically, assuming a foundation model has been trained on natural images, this would be transferring to medical imaging, video, audio and test contexts. This would shed light on the performance of a model's middle layers.

2. **Task shifting** contexts: Contexts where a model is tasked with performing a previously unseen task, for example, transferring from classification to segmentation or relational reasoning.

3. **Domain shifting** contexts: Contexts where a model is required to perform a task on a domain that is different from the one it was trained on. For example moving from natural images on ImageNet at 224x224 resolution to black and white Omniglot characters at 28x28 resolution, or, moving from ImageNet to images of fungi. More extreme domain shifts would be going from natural images to medical images for example.

## H   TASK ADAPTER DETAILS

**Classification**: For classification after a given encoder's output features we apply a linear layer as is standard.

**Segmentation**: We extract features after every stage within an encoder, i.e. before each pooling layer in conv-net architectures, and, after a transformer block in ViT-derivative encoders. We then upscale those to 64x64 before we concatenate and feed to a transformer decoder.

**Relational Reasoning**: We process images with our designated encoder, and text with a CLIP text encoder. We then concatenate the features, and feed into a transformer that considers receives as tokens each feature map of the image encoder and each token in the CLIP text encoder output, therefore allowing relational associations between these to be learned. After the transformer, we take the mean of the output tokens and apply a linear layer.

**Few Shot Img Classification**: We use the encoders as they are and employ a prototypical network as our method of achieving few shot learning.

**Image to Text Zero Shot**: We use the standard CLIP cosine distance-based matching, and we employ BERT embeddings for text and for images we apply our chosen encoder.

**Video Classification and Regression**: We process each frame with our chosen encoder which produce one vector per frame, and then use a 4-layer transformer to process the temporal axis before we apply a linear layer mapping to our classes or our single value output.

# I  EXPERIMENTAL DETAILS

**Experimental Environment Details**: GPUs: 4 x A6000 Ada @ 48GB, CPUs: 128 Core AMD EPYC 7713 64-Core Processor, RAM: 1 TB, HD: 15TB NVME. All experiments were done with BF16 precision.

# J  ADDITIONAL RESULTS

## J.1  FULL DETAILS ON DISCOVERED COMBINATIONS

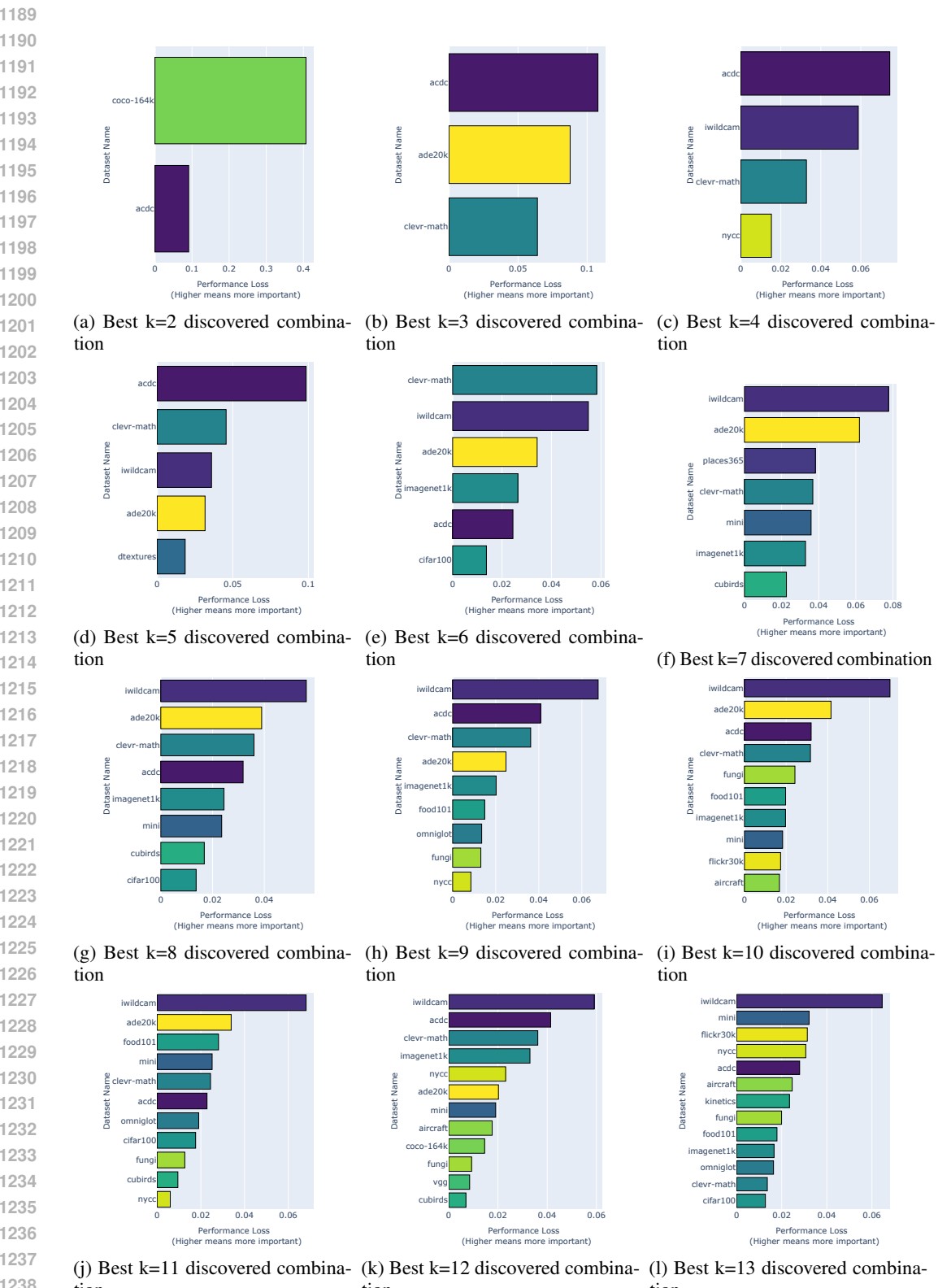

(a) Best k=2 discovered combination

(b) Best k=3 discovered combination

(c) Best k=4 discovered combination

(d) Best k=5 discovered combination

(e) Best k=6 discovered combination

(f) Best k=7 discovered combination

(g) Best k=8 discovered combination

(h) Best k=9 discovered combination

(i) Best k=10 discovered combination

(j) Best k=11 discovered combination

(k) Best k=12 discovered combination

(l) Best k=13 discovered combination

Figure 6: Degradation of predictive power when a given benchmark is removed and the meta-model trained from scratch, for different best combinations in varying $k$.

| Metric ↓ \| Model → | cvnxtv2 | siglip | clip | flex | deit | laion | vit | dino | smvit | rnx50 | effv2 | r50a1 | effrmr | seffv2 | sflex | svit | whspr | sr50a1 | bert | bart | mpnet |
|---|---|---|---|---|---|---|---|---|---|---|---|---|---|---|---|---|---|---|---|---|---|
| **Img Class** | | | | | | | | | | | | | | | | | | | | | |
| CIFAR-100 Acc@1 | 84.2 | 74.6 | 76.9 | 75.1 | 66.7 | 75.1 | 66.6 | 55.7 | 50.3 | 69.3 | 67.3 | 34.3 | 15.6 | 37.6 | 10.3 | 7.8 | 11.0 | 15.9 | 14.5 | 9.0 | 1.0 |
| CIFAR-100 Acc@5 | 97.4 | 93.8 | 95.1 | 94.4 | 90.9 | 93.9 | 89.7 | 83.6 | 80.1 | 91.9 | 90.7 | 65.9 | 42.3 | 67.6 | 30.6 | 25.5 | 31.6 | 40.2 | 38.1 | 29.2 | 5.0 |
| CIFAR-100 Loss | 0.6 | 0.9 | 0.8 | 0.9 | 1.2 | 0.9 | 1.2 | 1.6 | 1.9 | 1.2 | 1.3 | 2.5 | 3.5 | 2.4 | 3.9 | 4.1 | 3.9 | 3.6 | 3.7 | 4.0 | 4.6 |
| Food-101 Acc@1 | 92.9 | 91.6 | 93.3 | 89.1 | 87.3 | 91.4 | 86.5 | 84.8 | 75.7 | 86.1 | 86.4 | 69.4 | 61.6 | 36.5 | 24.5 | 25.8 | 17.0 | 16.3 | 18.7 | 11.6 | 8.5 |
| Food-101 Acc@5 | 99.0 | 98.7 | 99.1 | 98.1 | 97.8 | 98.7 | 97.4 | 97.0 | 93.5 | 97.2 | 97.1 | 91.0 | 86.6 | 66.1 | 51.0 | 52.8 | 41.1 | 38.9 | 43.0 | 32.2 | 26.1 |
| Food-101 Loss | 0.3 | 0.3 | 0.2 | 0.4 | 0.4 | 0.3 | 0.5 | 0.5 | 1.0 | 0.6 | 0.6 | 1.1 | 1.5 | 2.6 | 3.2 | 3.1 | 3.6 | 3.6 | 3.5 | 3.9 | 4.1 |
| HWhale Individual Acc@1 | 75.6 | 31.7 | 35.2 | 48.4 | 23.7 | 21.0 | 27.5 | 9.1 | 3.6 | 78.7 | 77.1 | 5.2 | 4.4 | 33.2 | 2.8 | 2.5 | 2.2 | 2.1 | 2.3 | 1.7 | 1.5 |
| HWhale Individual Acc@5 | 84.6 | 49.5 | 53.9 | 64.5 | 40.9 | 37.9 | 46.0 | 22.0 | 11.0 | 86.7 | 83.6 | 14.8 | 11.9 | 52.5 | 9.2 | 8.1 | 6.9 | 6.8 | 7.6 | 5.7 | 5.4 |
| HWhale Individual Loss | 1.6 | 4.6 | 4.3 | 3.6 | 4.9 | 5.1 | 4.7 | 5.9 | 6.7 | 1.3 | 1.5 | 6.4 | 6.6 | 3.9 | 7.0 | 7.1 | 7.3 | 7.3 | 7.2 | 7.5 | 7.4 |
| HWhale Species Acc@1 | 99.8 | 99.8 | 99.7 | 99.8 | 99.5 | 99.7 | 99.7 | 99.2 | 95.4 | 99.7 | 99.7 | 92.1 | 92.8 | 96.5 | 76.5 | 74.5 | 64.3 | 65.8 | 71.2 | 59.3 | 62.9 |
| HWhale Species Acc@5 | 100.0 | 100.0 | 100.0 | 100.0 | 100.0 | 100.0 | 100.0 | 99.9 | 99.6 | 100.0 | 100.0 | 98.9 | 99.1 | 99.8 | 96.1 | 95.8 | 92.0 | 92.6 | 94.2 | 89.8 | 91.1 |
| HWhale Species Loss | 0.0 | 0.0 | 0.0 | 0.0 | 0.0 | 0.0 | 0.0 | 0.0 | 0.2 | 0.0 | 0.0 | 0.3 | 0.2 | 0.1 | 0.8 | 0.8 | 1.2 | 1.1 | 0.9 | 1.4 | 1.2 |
| ImageNet-1K Acc@1 | 85.3 | 81.9 | 76.0 | 82.3 | 82.1 | 74.1 | 68.3 | 77.9 | 75.5 | 77.6 | 73.5 | 72.5 | 44.6 | 16.9 | 3.2 | 2.4 | 2.2 | 1.3 | 1.5 | 0.8 | 0.2 |
| ImageNet-1K Acc@5 | 96.8 | 95.8 | 93.7 | 95.5 | 94.7 | 93.1 | 89.1 | 93.0 | 90.8 | 93.3 | 91.4 | 90.5 | 72.5 | 37.3 | 10.1 | 8.2 | 7.7 | 4.7 | 5.2 | 3.2 | 1.2 |
| ImageNet-1K Loss | 0.6 | 0.8 | 1.0 | 0.8 | 0.8 | 1.1 | 1.3 | 1.0 | 2.3 | 1.0 | 1.2 | 1.1 | 2.8 | 4.3 | 6.0 | 6.1 | 6.1 | 6.5 | 6.4 | 6.6 | 6.8 |
| Places365 Acc@1 | 54.7 | 53.5 | 54.1 | 52.1 | 49.0 | 53.7 | 47.5 | 47.3 | 27.1 | 51.8 | 51.5 | 40.9 | 25.2 | 26.6 | 9.0 | 8.6 | 7.5 | 5.0 | 5.2 | 3.0 | 2.2 |
| Places365 Acc@5 | 85.3 | 84.1 | 84.7 | 83.3 | 80.8 | 84.3 | 79.9 | 79.5 | 59.9 | 82.9 | 82.6 | 73.5 | 55.2 | 55.5 | 26.3 | 25.0 | 22.4 | 16.4 | 16.4 | 11.0 | 9.0 |
| Places365 Loss | 1.7 | 1.7 | 1.7 | 1.8 | 1.9 | 1.7 | 2.0 | 2.0 | 3.1 | 1.8 | 1.8 | 2.3 | 3.3 | 3.2 | 4.5 | 4.6 | 4.6 | 5.0 | 5.0 | 5.3 | 5.3 |
| **Task Mean** | **88.0** | 79.6 | 80.1 | 81.9 | 76.1 | 76.9 | 74.8 | 70.8 | 63.5 | 84.6 | 83.4 | 62.4 | 51.0 | 52.2 | 29.1 | 28.1 | 25.5 | 25.5 | 26.5 | 21.4 | 17.8 |
| **Few-Shot Img Class** | | | | | | | | | | | | | | | | | | | | | |
| Aircraft Acc@1 | 96.7 | 96.6 | 97.4 | 95.9 | 95.3 | 96.7 | 96.3 | 94.4 | 92.9 | 91.6 | 90.6 | 86.2 | 78.2 | 59.2 | 54.9 | 50.4 | 55.1 | 58.2 | 61.2 | 60.8 | 57.2 |
| Aircraft Loss | 0.2 | 0.2 | 0.2 | 0.2 | 0.3 | 0.2 | 0.2 | 0.3 | 0.3 | 0.4 | 1.2 | 0.4 | 311.5 | 44.1 | 2.1 | 2.1 | 1.6 | 2.3 | 2.5 | 1.2 | 1.6 |
| CUBirds Acc@1 | 98.0 | 97.9 | 97.2 | 96.4 | 96.2 | 96.6 | 95.9 | 94.4 | 93.4 | 92.8 | 92.1 | 89.4 | 86.3 | 52.5 | 50.0 | 45.2 | 44.4 | 31.9 | 48.4 | 50.3 | 48.5 |
| CUBirds Loss | 0.2 | 0.2 | 0.2 | 0.3 | 0.2 | 0.2 | 0.2 | 0.3 | 0.3 | 0.4 | 0.5 | 0.4 | 33.7 | 2.5 | 3.6 | 3.5 | 2.3 | 8.8 | 3.2 | 2.0 | 1.6 |
| DTextures Acc@1 | 85.0 | 85.2 | 88.6 | 78.9 | 81.9 | 86.1 | 80.8 | 79.4 | 81.9 | 77.7 | 60.3 | 77.2 | 68.5 | 46.6 | 50.2 | 50.5 | 50.0 | 33.1 | 44.6 | 49.8 | 38.3 |
| DTextures Loss | 0.9 | 0.7 | 0.5 | 1.1 | 0.9 | 0.7 | 1.1 | 1.2 | 0.9 | 0.7 | 14.3 | 0.6 | 3.6 | 1.8 | 2.5 | 2.7 | 2.4 | 5.0 | 2.0 | 1.9 | 1.4 |
| Fungi Acc@1 | 85.8 | 85.6 | 85.7 | 83.7 | 80.6 | 85.2 | 81.3 | 77.4 | 77.7 | 74.1 | 73.7 | 67.1 | 59.2 | 27.6 | 38.0 | 37.0 | 33.9 | 28.2 | 32.9 | 33.8 | 7.6 |
| Fungi Loss | 0.6 | 0.6 | 0.6 | 0.7 | 0.8 | 0.6 | 0.8 | 0.9 | 0.8 | 1.1 | 5.8 | 1.1 | 1031.2 | 2.6 | 2.2 | 2.2 | 2.2 | 2.4 | 2.4 | 2.3 | 2.9 |
| Mini-Imagenet Acc@1 | 97.0 | 96.2 | 93.1 | 99.1 | 98.8 | 90.8 | 89.9 | 98.7 | 92.9 | 94.1 | 63.2 | 93.2 | 90.9 | 36.7 | 45.9 | 47.2 | 44.8 | 34.2 | 39.7 | 37.3 | 36.8 |
| Mini-Imagenet Loss | 0.1 | 0.1 | 0.3 | 0.0 | 0.0 | 0.3 | 0.4 | 0.1 | 0.2 | 0.3 | 23.7 | 0.3 | 0.6 | 2.4 | 1.6 | 1.6 | 1.6 | 2.1 | 1.8 | 1.9 | 1.9 |
| Omniglot Acc@1 | 98.6 | 98.9 | 99.0 | 98.9 | 98.7 | 98.9 | 98.8 | 98.6 | 98.6 | 98.5 | 98.7 | 95.5 | 95.8 | 98.2 | 93.4 | 93.6 | 82.9 | 80.5 | 90.2 | 84.1 | 90.7 |
| Omniglot Loss | 0.1 | 0.1 | 0.1 | 0.1 | 0.1 | 0.1 | 0.1 | 0.1 | 0.1 | 0.1 | 0.1 | 0.2 | 0.2 | 0.1 | 0.3 | 0.2 | 0.6 | 0.7 | 0.4 | 0.6 | 0.3 |
| VGG Flowers Acc@1 | 99.7 | 98.9 | 98.6 | 96.7 | 96.2 | 97.0 | 95.9 | 95.5 | 93.4 | 87.9 | 91.3 | 89.3 | 90.6 | 59.6 | 69.4 | 69.4 | 63.0 | 53.4 | 59.1 | 59.4 | 60.8 |
| VGG Flowers Loss | 0.1 | 0.1 | 0.1 | 0.2 | 0.2 | 0.1 | 0.2 | 0.2 | 0.2 | 0.4 | 0.5 | 0.4 | 0.3 | 1.6 | 1.8 | 1.6 | 1.4 | 4.2 | 2.5 | 1.6 | 1.5 |
| **Task Mean** | 94.4 | 94.2 | 94.2 | 92.8 | 92.5 | 93.1 | 91.3 | 91.2 | 90.1 | 88.1 | 81.4 | 85.4 | 81.4 | 54.3 | 57.4 | 56.2 | 53.4 | 45.6 | 53.7 | 53.6 | 48.6 |
| **Img Seg** | | | | | | | | | | | | | | | | | | | | | |
| ADE20K CE Loss | 1.1 | 1.0 | 1.1 | 1.1 | 1.3 | 1.0 | 1.3 | 1.4 | 1.7 | 2.0 | 2.2 | 2.8 | 2.8 | 3.3 | 3.8 | 3.8 | 3.7 | 3.7 | 3.7 | 3.7 | 3.8 |
| ADE20K Focal Loss | 0.2 | 0.2 | 0.2 | 0.2 | 0.2 | 0.2 | 0.3 | 0.3 | 0.3 | 0.4 | 0.5 | 0.6 | 0.6 | 0.8 | 0.9 | 0.9 | 0.9 | 0.9 | 0.9 | 0.9 | 0.9 |
| ADE20K Mean Acc@ | 59.8 | 60.8 | 57.5 | 56.0 | 49.1 | 57.3 | 44.2 | 45.1 | 36.3 | 26.8 | 20.4 | 17.9 | 15.2 | 3.6 | 1.6 | 1.6 | 1.8 | 1.8 | 1.8 | 1.8 | 1.8 |
| ADE20K Overall Acc@ | 71.8 | 74.4 | 72.6 | 71.4 | 66.9 | 72.4 | 64.2 | 63.5 | 57.5 | 49.6 | 43.9 | 34.6 | 39.7 | 21.3 | 11.7 | 11.9 | 13.1 | 14.1 | 14.0 | 14.4 | 14.2 |
| ADE20K mIoU | 46.8 | 47.1 | 44.0 | 43.7 | 37.8 | 43.4 | 33.2 | 33.3 | 25.9 | 18.2 | 14.2 | 11.7 | 9.8 | 1.5 | 0.5 | 0.4 | 0.6 | 0.4 | 0.4 | 0.5 | 0.4 |
| Cityscapes CE Loss | 0.2 | 0.2 | 0.2 | 0.2 | 0.2 | 0.2 | 0.2 | 0.2 | 0.2 | 0.4 | 0.2 | 0.4 | 4.1 | 0.3 | 0.7 | 0.7 | 0.9 | 0.9 | 3.9 | 4.0 | 3.8 |
| Cityscapes Focal Loss | 0.0 | 0.0 | 0.0 | 0.0 | 0.0 | 0.0 | 0.0 | 0.0 | 0.0 | 0.1 | 0.0 | 0.1 | 1.0 | 0.1 | 0.1 | 0.1 | 0.1 | 0.1 | 0.9 | 0.9 | 0.9 |
| Cityscapes Overall Acc@ | 92.5 | 94.2 | 93.9 | 93.6 | 93.1 | 93.7 | 93.4 | 93.1 | 92.8 | 88.5 | 93.2 | 87.4 | 41.5 | 90.4 | 78.1 | 78.6 | 72.2 | 75.4 | 47.4 | 37.7 | 47.3 |
| Cityscapes mIoU | 62.3 | 69.8 | 67.6 | 67.5 | 63.9 | 67.7 | 63.9 | 61.4 | 59.5 | 40.8 | 64.2 | 40.2 | 2.5 | 46.7 | 22.8 | 23.5 | 17.1 | 18.6 | 2.7 | 2.0 | 2.7 |
| COCO-10K CE Loss | 3.0 | 1.3 | 1.5 | 1.4 | 1.5 | 1.4 | 1.5 | 1.6 | 1.6 | 2.1 | 2.6 | 3.3 | 3.5 | 3.6 | 4.5 | 3.8 | 4.0 | 3.6 | 4.1 | 3.7 | 4.1 |
| COCO-10K Focal Loss | 0.7 | 0.3 | 0.3 | 0.3 | 0.3 | 0.3 | 0.3 | 0.3 | 0.3 | 0.4 | 0.6 | 0.6 | 0.8 | 0.8 | 1.1 | 0.9 | 0.9 | 0.8 | 1.0 | 0.9 | 1.0 |
| COCO-10K Mean Acc@ | 38.8 | 50.6 | 47.2 | 46.0 | 43.4 | 44.9 | 41.2 | 43.5 | 40.7 | 27.0 | 15.8 | 8.2 | 20.9 | 2.2 | 1.7 | 1.9 | 1.3 | 2.9 | 0.6 | 2.5 | 0.6 |
| COCO-10K Overall Acc@ | 57.9 | 69.8 | 66.4 | 66.0 | 64.4 | 65.9 | 62.8 | 63.1 | 61.2 | 51.3 | 40.1 | 23.3 | 45.2 | 20.9 | 15.2 | 20.5 | 14.7 | 24.6 | 9.4 | 22.5 | 9.3 |
| COCO-10K mIoU | 26.9 | 39.5 | 35.6 | 35.1 | 32.8 | 33.6 | 29.8 | 31.0 | 28.6 | 18.4 | 10.2 | 5.7 | 14.0 | 1.1 | 0.9 | 0.8 | 0.4 | 1.6 | 0.1 | 1.3 | 0.1 |
| COCO-164K CE Loss | 1.9 | 1.4 | 1.5 | 1.5 | 1.6 | 1.5 | 1.6 | 1.7 | 1.8 | 2.2 | 2.7 | 3.5 | 7.0 | 3.7 | 4.3 | 3.9 | 4.0 | 4.0 | 4.2 | 3.7 | 4.2 |
| COCO-164K Focal Loss | 0.4 | 0.3 | 0.3 | 0.3 | 0.4 | 0.3 | 0.4 | 0.4 | 0.4 | 0.5 | 0.6 | 0.8 | 1.7 | 0.9 | 1.0 | 0.9 | 0.9 | 0.9 | 1.0 | 0.9 | 1.0 |
| COCO-164K Mean Acc@ | 45.9 | 50.1 | 46.9 | 45.3 | 42.6 | 44.5 | 38.6 | 43.0 | 38.7 | 25.4 | 14.7 | 7.0 | 21.3 | 2.0 | 1.5 | 1.9 | 1.5 | 1.8 | 0.6 | 2.5 | 0.7 |
| COCO-164K Overall Acc@ | 60.9 | 65.8 | 63.5 | 63.0 | 60.3 | 63.2 | 59.5 | 59.1 | 55.6 | 47.9 | 39.3 | 20.3 | 39.3 | 19.2 | 13.6 | 19.4 | 15.6 | 18.3 | 9.5 | 21.7 | 9.6 |
| COCO-164K mIoU | 32.7 | 36.7 | 33.8 | 33.0 | 30.5 | 32.4 | 27.0 | 28.9 | 25.7 | 16.8 | 9.7 | 4.7 | 13.7 | 1.0 | 0.7 | 0.7 | 0.5 | 0.7 | 0.1 | 1.1 | 0.1 |
| NYU CE Loss | 2.5 | 1.5 | 2.0 | 2.3 | 1.5 | 2.5 | 2.3 | 1.5 | 1.6 | 1.6 | 1.8 | 1.6 | 1.4 | 1.6 | 1.6 | 1.6 | 1.7 | 1.5 | 1.5 | 1.5 | 1.5 |
| NYU Dice Score | 0.8 | 0.8 | 0.8 | 0.8 | 0.8 | 0.8 | 0.8 | 0.8 | 0.8 | 0.8 | 0.8 | 0.8 | 0.8 | 0.8 | 0.8 | 0.8 | 0.7 | 0.8 | 0.8 | 0.8 | 0.8 |
| NYU Focal Loss | 0.5 | 0.2 | 0.4 | 0.5 | 0.3 | 0.5 | 0.5 | 0.3 | 0.3 | 0.3 | 0.3 | 0.3 | 0.2 | 0.3 | 0.3 | 0.3 | 0.3 | 0.2 | 0.2 | 0.2 | 0.2 |
| NYU Mean Acc@ | 19.7 | 21.5 | 13.0 | 19.6 | 22.7 | 19.4 | 19.7 | 23.0 | 22.9 | 18.5 | 18.3 | 12.7 | 18.9 | 14.1 | 10.0 | 10.1 | 10.2 | 13.0 | 11.9 | 11.7 | 12.0 |
| NYU Overall Acc@ | 19.0 | 37.2 | 30.8 | 30.0 | 42.8 | 25.2 | 27.3 | 34.7 | 31.2 | 33.4 | 30.7 | 33.4 | 39.1 | 31.9 | 34.6 | 34.6 | 34.3 | 36.3 | 37.2 | 37.1 | 37.4 |
| NYU mIoU | 7.5 | 7.7 | 7.8 | 6.9 | 12.2 | 5.7 | 6.1 | 12.1 | 11.0 | 5.9 | 8.3 | 6.4 | 10.5 | 6.8 | 3.5 | 3.7 | 2.9 | 7.2 | 5.4 | 5.0 | 5.4 |
| Pascal CE Loss | 1.0 | 0.5 | 0.5 | 0.6 | 0.9 | 0.5 | 0.8 | 0.8 | 0.9 | 1.4 | 1.5 | 2.2 | 3.1 | 2.3 | 2.4 | 2.4 | 2.4 | 2.4 | 2.6 | 2.5 | 2.6 |
| Pascal Dice Loss | 0.8 | 0.6 | 0.4 | 0.5 | 0.5 | 0.4 | 0.5 | 0.5 | 0.4 | 0.5 | 0.4 | 0.5 | 0.5 | 0.2 | 0.4 | 0.4 | 0.4 | 0.4 | 0.5 | 0.5 | 0.4 |
| Pascal Focal Loss | 0.2 | 0.1 | 0.1 | 0.1 | 0.2 | 0.1 | 0.2 | 0.1 | 0.2 | 0.2 | 0.3 | 0.4 | 0.7 | 0.5 | 0.5 | 0.5 | 0.5 | 0.5 | 0.5 | 0.5 | 0.5 |
| Pascal Loss | 1.4 | 0.5 | 0.1 | 0.3 | 0.3 | 0.4 | 0.3 | 0.6 | 0.6 | 0.5 | 0.4 | 1.4 | 4.2 | 1.6 | 1.6 | 1.6 | 1.6 | 1.6 | 3.4 | 1.7 | 3.5 |
| Pascal Mean Acc@ | 42.2 | 43.5 | 44.2 | 39.6 | 38.8 | 37.4 | 34.7 | 40.3 | 29.1 | 20.7 | 16.2 | 10.6 | 18.0 | 3.5 | 3.1 | 2.8 | 3.3 | 4.5 | 2.6 | 3.3 | 2.5 |
| Pascal Overall Acc@ | 75.1 | 87.6 | 87.2 | 86.6 | 77.5 | 86.6 | 78.9 | 79.5 | 76.7 | 68.2 | 60.6 | 49.7 | 66.6 | 37.3 | 34.2 | 35.4 | 37.4 | 39.6 | 34.4 | 35.3 | 32.3 |
| Pascal mIoU | 32.8 | 34.8 | 35.7 | 30.6 | 31.4 | 28.3 | 27.5 | 29.8 | 24.0 | 16.6 | 11.7 | 6.8 | 14.0 | 1.7 | 1.3 | 1.1 | 1.4 | 2.3 | 1.0 | 1.4 | 0.9 |
| **Task Mean** | 44.1 | 49.6 | 47.1 | 46.4 | 45.1 | 45.7 | 41.8 | 43.6 | 39.9 | 31.9 | 28.5 | 21.2 | 24.0 | 17.0 | 13.1 | 13.9 | 12.7 | 14.7 | 10.0 | 11.2 | 9.9 |
| **Img Relational** | | | | | | | | | | | | | | | | | | | | | |
| CLEVR Acc@1 | 52.5 | 52.7 | 52.7 | 52.1 | 52.6 | 52.6 | 52.8 | 52.8 | 51.6 | 50.1 | 40.6 | 49.3 | 45.2 | 39.3 | 46.1 | 45.9 | 46.4 | 44.9 | 42.6 | 42.5 | 41.2 |
| CLEVR Colour Acc@1 | 35.4 | 36.1 | 36.4 | 35.0 | 35.5 | 35.6 | 35.3 | 36.1 | 34.2 | 26.8 | 15.7 | 24.7 | 14.7 | 12.5 | 25.7 | 29.4 | 28.8 | 22.8 | 13.2 | 13.0 | 13.2 |
| CLEVR Colour Loss | 1.5 | 1.5 | 1.5 | 1.5 | 1.5 | 1.5 | 1.5 | 1.5 | 1.6 | 1.9 | 2.1 | 2.0 | 2.1 | 2.1 | 2.0 | 1.9 | 1.9 | 2.0 | 2.1 | 2.1 | 2.1 |
| CLEVR Count Acc@1 | 45.8 | 45.8 | 45.8 | 45.9 | 45.8 | 45.7 | 45.7 | 45.6 | 45.6 | 45.3 | 39.0 | 45.1 | 44.8 | 37.9 | 45.1 | 44.7 | 44.8 | 44.9 | 44.7 | 44.7 | 43.0 |
| CLEVR Count Loss | 1.1 | 1.2 | 1.1 | 1.2 | 1.2 | 1.2 | 1.2 | 1.2 | 1.2 | 1.2 | 1.3 | 1.2 | 1.2 | 1.4 | 1.2 | 1.2 | 1.2 | 1.2 | 1.2 | 1.2 | 1.2 |
| CLEVR Material Acc@1 | 60.5 | 60.6 | 60.5 | 60.0 | 60.5 | 60.6 | 61.4 | 61.3 | 60.2 | 58.6 | 52.1 | 57.5 | 53.7 | 49.8 | 53.7 | 51.7 | 54.0 | 53.0 | 49.8 | 50.5 | 49.9 |
| CLEVR Material Loss | 0.7 | 0.7 | 0.7 | 0.7 | 0.7 | 0.7 | 0.6 | 0.6 | 0.7 | 0.7 | 0.7 | 0.7 | 0.7 | 0.7 | 0.7 | 0.7 | 0.7 | 0.7 | 0.7 | 0.7 | 0.7 |
| CLEVR Shape Acc@1 | 52.1 | 52.4 | 52.5 | 51.1 | 52.2 | 52.4 | 52.9 | 51.2 | 49.9 | 50.2 | 34.3 | 50.2 | 44.8 | 33.3 | 35.8 | 34.9 | 36.1 | 34.6 | 34.6 | 33.7 | 33.4 |
| CLEVR Shape Loss | 0.9 | 0.9 | 0.9 | 1.0 | 0.9 | 0.9 | 0.9 | 1.0 | 1.0 | 1.1 | 1.1 | 1.1 | 1.1 | 1.1 | 1.1 | 1.1 | 1.1 | 1.1 | 1.1 | 1.1 | 1.1 |
| CLEVR Size Acc@1 | 61.0 | 61.1 | 61.3 | 60.7 | 61.1 | 60.8 | 62.0 | 62.3 | 60.9 | 59.6 | 53.5 | 58.3 | 55.7 | 50.6 | 56.2 | 55.2 | 55.2 | 54.6 | 54.2 | 54.1 | 50.1 |
| CLEVR Size Loss | 0.6 | 0.6 | 0.6 | 0.6 | 0.6 | 0.6 | 0.6 | 0.6 | 0.6 | 0.7 | 0.7 | 0.7 | 0.7 | 0.7 | 0.7 | 0.7 | 0.7 | 0.7 | 0.7 | 0.7 | 0.7 |
| CLEVR Yes/No Acc@1 | 60.7 | 60.5 | 60.8 | 60.6 | 60.5 | 60.7 | 60.4 | 60.4 | 60.2 | 59.8 | 53.3 | 59.9 | 59.6 | 51.4 | 60.1 | 59.2 | 59.5 | 59.8 | 59.5 | 59.3 | 58.6 |
| CLEVR Yes/No Loss | 0.6 | 0.6 | 0.6 | 0.6 | 0.6 | 0.6 | 0.6 | 0.6 | 0.6 | 0.6 | 0.7 | 0.6 | 0.6 | 0.7 | 0.6 | 0.6 | 0.6 | 0.6 | 0.6 | 0.6 | 0.6 |
| CLEVR-Math Acc@1 | 79.3 | 65.9 | 68.8 | 59.9 | 73.7 | 62.9 | 60.5 | 59.3 | 58.3 | 55.6 | 44.0 | 56.0 | 56.6 | 30.2 | 46.9 | 46.6 | 46.2 | 45.7 | 44.8 | 42.1 | 36.4 |
| CLEVR-Math Acc@5 | 99.8 | 99.5 | 99.6 | 98.9 | 99.7 | 99.3 | 99.2 | 98.9 | 98.9 | 98.8 | 97.7 | 98.8 | 98.8 | 86.1 | 98.1 | 98.1 | 98.1 | 97.7 | 97.5 | 96.9 | 92.8 |
| CLEVR-Math Loss | 0.5 | 0.8 | 0.7 | 0.9 | 0.6 | 0.8 | 0.9 | 0.9 | 1.0 | 1.0 | 1.3 | 1.0 | 1.0 | 1.7 | 1.2 | 1.2 | 1.2 | 1.2 | 1.3 | 1.3 | 1.5 |
| **Task Mean** | 60.8 | 59.4 | 59.8 | 58.2 | 60.2 | 59.0 | 58.9 | 58.7 | 57.8 | 56.1 | 47.8 | 55.5 | 52.7 | 43.5 | 52.0 | 51.7 | 52.1 | 50.9 | 49.0 | 48.5 | 46.5 |
| **Medical Class** | | | | | | | | | | | | | | | | | | | | | |
| Chexpert 0 APS | 75.7 | 76.5 | 76.6 | 76.8 | 76.8 | 74.7 | 76.0 | 75.8 | 76.3 | 75.1 | 75.2 | 69.1 | 70.3 | 65.3 | 20.6 | 22.3 | 21.9 | 29.4 | 31.6 | 25.2 | 23.2 |

| | | | | | | | | | | | | | | | | | | | | | |
|---|---|---|---|---|---|---|---|---|---|---|---|---|---|---|---|---|---|---|---|---|---|
| Chexpert 0 AUC | 91.3 | 92.1 | 92.5 | 92.3 | 92.6 | 91.4 | 92.2 | 92.3 | 92.6 | 91.0 | 91.6 | 89.9 | 90.5 | 88.5 | 61.5 | 64.0 | 65.2 | 71.3 | 72.3 | 66.4 | 65.9 |
| Chexpert 0 BS | 7.8 | 7.4 | 7.3 | 7.4 | 7.0 | 7.5 | 7.3 | 7.3 | 6.9 | 7.9 | 7.3 | 7.9 | 7.7 | 8.4 | 12.6 | 12.5 | 12.5 | 11.9 | 12.1 | 12.1 | 12.4 |
| Chexpert 1 APS | 55.3 | 55.2 | 55.5 | 55.8 | 54.2 | 54.4 | 54.2 | 52.1 | 55.9 | 53.1 | 53.3 | 44.2 | 43.0 | 33.5 | 28.9 | 30.1 | 31.0 | 28.9 | 30.1 | 29.9 | 28.5 |
| Chexpert 1 AUC | 75.7 | 76.0 | 75.3 | 77.0 | 75.4 | 75.3 | 75.3 | 73.8 | 76.1 | 74.9 | 75.2 | 69.4 | 69.8 | 64.1 | 56.3 | 56.9 | 57.7 | 57.1 | 57.6 | 57.5 | 57.0 |
| Chexpert 1 BS | 18.8 | 18.5 | 19.4 | 20.2 | 18.7 | 20.6 | 20.6 | 18.6 | 16.3 | 17.6 | 20.3 | 17.2 | 17.2 | 18.3 | 18.7 | 18.6 | 18.6 | 18.6 | 18.5 | 18.6 | 18.6 |
| Chexpert 2 APS | 43.8 | 43.8 | 43.5 | 45.1 | 45.5 | 44.8 | 43.9 | 42.3 | 43.6 | 43.5 | 44.4 | 41.8 | 42.8 | 35.3 | 30.1 | 31.1 | 30.4 | 32.3 | 32.6 | 31.2 | 30.8 |
| Chexpert 2 AUC | 71.8 | 71.2 | 71.8 | 72.4 | 72.1 | 72.0 | 71.7 | 71.3 | 71.7 | 70.5 | 71.1 | 69.9 | 70.9 | 63.1 | 58.6 | 59.0 | 58.7 | 60.7 | 60.5 | 60.1 | 58.9 |
| Chexpert 2 BS | 18.5 | 17.8 | 21.0 | 21.1 | 18.6 | 21.2 | 20.5 | 19.1 | 17.0 | 16.0 | 20.4 | 16.2 | 16.2 | 17.4 | 18.4 | 18.2 | 18.1 | 17.9 | 17.9 | 17.8 | 17.9 |
| Chexpert 3 APS | 80.7 | 80.9 | 80.8 | 82.1 | 81.7 | 80.5 | 79.7 | 78.6 | 79.1 | 79.2 | 80.6 | 73.5 | 75.3 | 58.6 | 51.7 | 50.3 | 52.4 | 53.2 | 54.0 | 48.8 | 49.4 |
| Chexpert 3 AUC | 86.8 | 86.8 | 86.5 | 87.9 | 87.2 | 86.6 | 85.9 | 84.6 | 85.8 | 84.9 | 87.0 | 82.3 | 83.5 | 73.0 | 65.6 | 65.5 | 65.2 | 65.6 | 67.2 | 64.2 | 64.2 |
| Chexpert 3 BS | 17.4 | 16.4 | 16.4 | 15.6 | 15.3 | 16.2 | 16.9 | 17.2 | 15.9 | 17.6 | 16.3 | 18.1 | 17.1 | 23.5 | 26.1 | 26.0 | 26.0 | 25.2 | 24.8 | 26.1 | 26.1 |
| Chexpert 4 APS | 53.4 | 49.5 | 50.1 | 53.4 | 54.5 | 50.9 | 52.6 | 50.8 | 52.3 | 49.9 | 50.7 | 41.7 | 44.9 | 47.3 | 38.4 | 36.7 | 36.0 | 39.2 | 37.9 | 35.9 | 33.2 |
| Chexpert 4 AUC | 87.5 | 86.7 | 87.0 | 88.1 | 88.0 | 87.0 | 87.3 | 86.8 | 87.7 | 86.0 | 86.4 | 84.1 | 85.1 | 84.8 | 81.7 | 80.3 | 80.8 | 81.5 | 81.3 | 79.4 | 79.3 |
| Chexpert 4 BS | 10.4 | 10.0 | 10.9 | 10.2 | 9.1 | 10.9 | 10.2 | 9.9 | 8.8 | 9.4 | 11.6 | 10.1 | 9.6 | 9.4 | 9.7 | 10.0 | 9.9 | 9.9 | 10.7 | 10.1 | 10.4 |
| Chexpert APS Macro | 61.6 | 61.0 | 61.2 | 62.6 | 62.3 | 60.9 | 61.2 | 59.9 | 61.5 | 59.8 | 60.2 | 54.1 | 55.2 | 48.0 | 33.9 | 34.1 | 34.3 | 35.7 | 36.9 | 33.7 | 33.0 |
| Chexpert AUC Macro | 82.5 | 82.5 | 82.3 | 83.2 | 82.9 | 82.5 | 82.4 | 81.8 | 82.8 | 81.1 | 81.9 | 79.1 | 79.9 | 74.7 | 64.7 | 65.1 | 65.5 | 67.0 | 67.6 | 65.3 | 64.9 |
| Chexpert BS Macro | 15.7 | 15.6 | 15.5 | 14.9 | 13.8 | 15.4 | 15.1 | 14.4 | 13.0 | 13.7 | 15.2 | 13.9 | 13.6 | 15.4 | 17.1 | 17.1 | 17.0 | 16.9 | 16.9 | 17.2 | 17.2 |
| Chexpert Loss | 0.3 | 0.4 | 0.5 | 0.3 | 0.3 | 0.3 | 0.4 | 0.4 | 0.3 | 0.3 | 0.4 | 0.3 | 0.4 | 0.4 | 0.5 | 0.5 | 0.5 | 0.5 | 0.4 | 0.5 | 0.5 |
| Diabetic 0 APS | 93.0 | 91.8 | 91.5 | 91.3 | 90.9 | 91.3 | 90.6 | 90.4 | 88.3 | 90.8 | 91.5 | 85.4 | 87.2 | 75.5 | 76.3 | 75.6 | 77.4 | 79.8 | 79.4 | 76.4 | 77.2 |
| Diabetic 0 AUC | 86.3 | 84.6 | 84.0 | 83.9 | 83.0 | 83.6 | 81.7 | 80.9 | 77.2 | 83.9 | 84.3 | 72.2 | 75.1 | 52.4 | 54.3 | 53.6 | 56.5 | 60.4 | 58.6 | 54.7 | 55.3 |
| Diabetic 0 BS | 10.7 | 11.9 | 12.3 | 12.4 | 12.6 | 12.6 | 13.0 | 13.0 | 14.6 | 12.1 | 11.7 | 16.5 | 15.7 | 19.0 | 19.5 | 19.4 | 19.3 | 19.1 | 18.6 | 19.0 | 19.0 |
| Diabetic 1 APS | 14.0 | 13.6 | 14.0 | 13.0 | 13.0 | 12.9 | 14.5 | 10.8 | 9.0 | 12.6 | 13.5 | 8.4 | 9.0 | 7.2 | 8.4 | 8.8 | 8.9 | 8.4 | 7.7 | 7.4 | 7.3 |
| Diabetic 1 AUC | 69.6 | 67.2 | 67.4 | 66.0 | 65.3 | 66.1 | 66.5 | 65.3 | 59.7 | 66.5 | 66.4 | 54.4 | 59.5 | 51.4 | 54.9 | 56.9 | 54.5 | 53.9 | 54.9 | 52.1 | 53.3 |
| Diabetic 1 BS | 6.1 | 6.4 | 6.5 | 6.1 | 6.0 | 6.8 | 6.4 | 5.8 | 5.8 | 6.0 | 6.4 | 6.9 | 5.3 | 6.5 | 6.7 | 6.5 | 6.9 | 6.4 | 6.3 | 6.4 | 6.3 |
| Diabetic 2 APS | 65.5 | 61.6 | 60.7 | 61.4 | 58.4 | 57.1 | 54.2 | 51.1 | 44.3 | 59.7 | 63.1 | 28.9 | 32.2 | 14.6 | 17.0 | 16.7 | 17.9 | 20.2 | 17.8 | 17.0 | 17.3 |
| Diabetic 2 AUC | 88.5 | 86.9 | 86.3 | 86.0 | 84.7 | 85.3 | 84.3 | 82.5 | 79.6 | 85.5 | 87.4 | 71.6 | 73.8 | 50.9 | 53.4 | 52.2 | 55.8 | 61.2 | 57.7 | 54.1 | 55.5 |
| Diabetic 2 BS | 8.0 | 8.5 | 9.0 | 9.3 | 9.7 | 9.0 | 9.5 | 9.9 | 10.7 | 9.2 | 8.3 | 11.7 | 11.7 | 12.1 | 12.7 | 12.8 | 12.6 | 12.7 | 11.9 | 12.4 | 12.5 |
| Diabetic 3 APS | 41.6 | 49.7 | 47.6 | 48.4 | 45.3 | 53.1 | 46.5 | 38.8 | 37.1 | 47.2 | 50.7 | 22.4 | 32.0 | 2.8 | 3.1 | 3.1 | 4.1 | 4.6 | 4.0 | 3.4 | 2.6 |
| Diabetic 3 AUC | 94.8 | 96.5 | 95.7 | 95.6 | 93.9 | 95.1 | 95.0 | 94.1 | 93.5 | 95.1 | 96.2 | 87.2 | 92.3 | 56.0 | 56.1 | 57.2 | 59.1 | 64.2 | 64.9 | 58.4 | 52.3 |
| Diabetic 3 BS | 1.9 | 1.6 | 1.6 | 1.6 | 1.7 | 1.9 | 1.7 | 1.8 | 1.8 | 1.7 | 1.5 | 2.0 | 2.1 | 2.4 | 2.4 | 2.5 | 2.3 | 2.2 | 2.3 | 2.1 | 2.1 |
| Diabetic 4 APS | 73.9 | 74.3 | 73.0 | 75.3 | 67.5 | 68.7 | 70.2 | 72.3 | 47.5 | 67.5 | 74.6 | 32.4 | 23.7 | 2.9 | 3.1 | 3.0 | 4.4 | 3.9 | 3.7 | 2.5 | 2.6 |
| Diabetic 4 AUC | 98.7 | 98.2 | 97.7 | 98.7 | 98.0 | 97.4 | 98.4 | 98.3 | 96.9 | 97.2 | 97.9 | 94.7 | 94.3 | 56.4 | 60.1 | 58.6 | 63.0 | 68.1 | 64.3 | 56.9 | 57.8 |
| Diabetic 4 BS | 1.0 | 1.1 | 1.0 | 0.9 | 1.1 | 1.1 | 0.9 | 0.9 | 1.3 | 1.1 | 0.8 | 1.4 | 1.8 | 1.9 | 1.9 | 1.8 | 1.7 | 1.7 | 1.8 | 1.9 | 1.8 |
| Diabetic APS Macro | 56.9 | 57.2 | 56.4 | 56.3 | 54.2 | 56.4 | 54.4 | 51.9 | 45.2 | 55.6 | 58.7 | 35.5 | 36.6 | 20.6 | 21.6 | 21.5 | 22.5 | 23.3 | 22.4 | 21.2 | 21.3 |
| Diabetic AUC Macro | 87.5 | 86.7 | 86.0 | 85.7 | 85.0 | 85.3 | 84.7 | 83.8 | 81.2 | 85.6 | 86.1 | 76.0 | 79.0 | 53.4 | 55.7 | 55.7 | 57.8 | 61.3 | 59.4 | 55.1 | 54.0 |
| Diabetic BS Macro | 5.5 | 6.0 | 6.1 | 6.1 | 6.2 | 6.4 | 6.3 | 6.4 | 7.0 | 6.1 | 5.8 | 7.7 | 7.4 | 8.4 | 8.7 | 8.6 | 8.6 | 8.5 | 8.2 | 8.4 | 8.4 |
| Diabetic Loss | 0.2 | 0.1 | 0.2 | 0.1 | 0.1 | 0.1 | 0.2 | 0.2 | 0.2 | 0.2 | 0.2 | 0.2 | 0.3 | 0.2 | 0.3 | 0.3 | 0.3 | 0.3 | 0.3 | 0.2 | 0.3 |
| HAM10K 0 APS | 94.3 | 90.0 | 90.3 | 88.7 | 89.2 | 90.9 | 89.8 | 89.0 | 83.3 | 88.2 | 84.1 | 47.4 | 58.0 | 30.4 | 32.8 | 25.8 | 25.0 | 41.2 | 46.2 | 34.4 | 33.8 |
| HAM10K 0 AUC | 99.1 | 98.2 | 98.3 | 97.6 | 97.8 | 98.2 | 97.7 | 98.0 | 96.7 | 97.6 | 97.0 | 89.0 | 91.7 | 80.6 | 81.2 | 78.5 | 79.4 | 85.2 | 86.9 | 82.1 | 79.7 |
| HAM10K 0 BS | 2.1 | 2.9 | 3.5 | 2.8 | 3.1 | 3.1 | 3.1 | 3.4 | 3.8 | 3.4 | 4.0 | 7.0 | 6.3 | 8.2 | 8.1 | 8.5 | 8.6 | 7.6 | 7.3 | 8.1 | 8.3 |
| HAM10K 1 APS | 99.2 | 99.2 | 99.1 | 99.2 | 99.2 | 99.1 | 99.1 | 99.2 | 98.7 | 98.9 | 98.1 | 96.2 | 96.5 | 94.2 | 93.9 | 93.7 | 93.1 | 95.5 | 96.0 | 94.0 | 93.7 |
| HAM10K 1 AUC | 98.9 | 98.7 | 98.4 | 98.5 | 98.4 | 98.4 | 98.4 | 98.4 | 97.3 | 98.1 | 97.1 | 92.7 | 93.5 | 89.7 | 88.7 | 88.1 | 87.8 | 91.0 | 91.9 | 88.3 | 87.3 |
| HAM10K 1 BS | 3.1 | 3.7 | 4.5 | 4.2 | 4.5 | 4.4 | 4.6 | 4.4 | 6.2 | 5.0 | 6.3 | 10.0 | 9.4 | 11.7 | 12.5 | 12.8 | 12.9 | 11.3 | 10.7 | 13.0 | 13.9 |
| HAM10K 2 APS | 95.5 | 98.6 | 89.0 | 94.4 | 88.7 | 92.1 | 92.4 | 95.3 | 69.7 | 81.6 | 89.0 | 11.3 | 5.0 | 5.7 | 5.2 | 8.1 | 2.2 | 19.5 | 12.2 | 7.4 | 3.6 |
| HAM10K 2 AUC | 99.9 | 100.0 | 99.7 | 99.9 | 99.7 | 99.8 | 99.8 | 99.9 | 99.3 | 98.4 | 99.8 | 81.1 | 75.6 | 79.4 | 79.6 | 73.0 | 68.2 | 90.8 | 87.2 | 81.2 | 78.3 |
| HAM10K 2 BS | 0.3 | 0.3 | 0.4 | 0.3 | 0.5 | 0.3 | 0.3 | 0.3 | 0.8 | 0.5 | 0.4 | 1.3 | 1.3 | 1.3 | 1.3 | 1.3 | 1.3 | 1.2 | 1.3 | 1.3 | 1.3 |
| HAM10K 3 APS | 88.0 | 85.5 | 83.9 | 85.2 | 86.2 | 83.0 | 84.0 | 82.5 | 74.2 | 80.8 | 74.3 | 41.9 | 46.7 | 34.7 | 35.1 | 33.2 | 31.5 | 42.5 | 48.4 | 42.4 | 35.2 |
| HAM10K 3 AUC | 96.7 | 95.5 | 95.6 | 95.9 | 96.1 | 95.3 | 95.9 | 96.1 | 94.4 | 95.4 | 92.5 | 83.8 | 84.9 | 81.7 | 80.0 | 80.3 | 80.7 | 85.9 | 88.1 | 84.2 | 82.6 |
| HAM10K 3 BS | 3.5 | 3.7 | 4.2 | 3.9 | 3.5 | 4.1 | 4.2 | 4.4 | 5.0 | 4.7 | 5.1 | 7.9 | 7.6 | 8.4 | 8.4 | 8.4 | 8.5 | 7.7 | 7.2 | 8.0 | 8.2 |
| HAM10K 4 APS | 99.5 | 100.0 | 99.7 | 98.2 | 100.0 | 98.5 | 100.0 | 98.5 | 98.7 | 96.4 | 96.9 | 26.8 | 21.9 | 33.6 | 32.3 | 24.6 | 26.4 | 52.8 | 73.8 | 34.8 | 11.5 |
| HAM10K 4 AUC | 100.0 | 100.0 | 100.0 | 100.0 | 100.0 | 100.0 | 100.0 | 100.0 | 100.0 | 99.9 | 100.0 | 92.3 | 94.5 | 84.2 | 89.0 | 89.5 | 87.1 | 97.7 | 97.6 | 92.0 | 78.7 |
| HAM10K 4 BS | 0.0 | 0.0 | 0.0 | 0.1 | 0.0 | 0.1 | 0.0 | 0.0 | 0.2 | 0.1 | 0.2 | 1.1 | 1.2 | 1.2 | 1.0 | 1.1 | 1.2 | 0.9 | 0.5 | 1.1 | 1.2 |
| HAM10K 5 APS | 95.6 | 94.8 | 94.5 | 91.5 | 90.3 | 93.7 | 91.6 | 90.8 | 83.2 | 88.0 | 91.4 | 54.1 | 67.0 | 41.8 | 36.4 | 36.8 | 22.3 | 48.1 | 41.3 | 36.6 | 26.8 |
| HAM10K 5 AUC | 99.7 | 99.7 | 99.6 | 99.5 | 99.2 | 99.5 | 99.4 | 99.0 | 98.8 | 98.0 | 99.4 | 94.7 | 96.6 | 92.9 | 92.0 | 91.5 | 87.5 | 94.0 | 92.5 | 90.3 | 88.2 |
| HAM10K 5 BS | 1.1 | 1.1 | 1.1 | 1.1 | 1.3 | 1.0 | 1.2 | 1.3 | 1.7 | 1.5 | 1.4 | 3.4 | 2.8 | 3.8 | 3.9 | 3.9 | 4.4 | 3.6 | 3.9 | 4.2 | 4.3 |
| HAM10K 6 APS | 89.2 | 85.2 | 83.9 | 86.3 | 88.0 | 87.6 | 84.7 | 83.3 | 75.8 | 81.3 | 83.2 | 28.4 | 33.4 | 31.4 | 30.5 | 29.5 | 24.5 | 39.6 | 36.6 | 25.8 | 23.5 |
| HAM10K 6 AUC | 99.3 | 98.3 | 99.1 | 98.6 | 98.9 | 99.1 | 99.3 | 98.6 | 98.0 | 98.0 | 98.6 | 91.6 | 93.6 | 91.2 | 91.9 | 91.4 | 89.1 | 92.3 | 93.4 | 90.5 | 88.5 |
| HAM10K 6 BS | 1.0 | 1.5 | 1.5 | 1.1 | 1.1 | 1.3 | 1.4 | 1.3 | 1.7 | 1.4 | 1.6 | 3.0 | 2.9 | 3.0 | 3.0 | 3.0 | 3.2 | 2.8 | 2.9 | 3.2 | 3.2 |
| HAM10K APS Macro | 94.5 | 93.3 | 91.4 | 92.2 | 91.3 | 92.1 | 91.6 | 90.8 | 83.4 | 87.9 | 87.1 | 43.7 | 46.9 | 38.8 | 38.0 | 35.9 | 32.2 | 48.5 | 50.6 | 37.6 | 32.6 |
| HAM10K AUC Macro | 99.1 | 98.6 | 98.7 | 98.5 | 98.6 | 98.6 | 98.7 | 98.5 | 97.8 | 97.9 | 97.5 | 89.3 | 90.1 | 85.6 | 86.1 | 84.6 | 82.8 | 91.0 | 91.1 | 85.9 | 83.3 |
| HAM10K BS Macro | 1.6 | 1.9 | 2.2 | 1.9 | 2.0 | 2.1 | 2.1 | 2.1 | 2.8 | 2.4 | 2.8 | 4.8 | 4.5 | 5.4 | 5.5 | 5.6 | 5.7 | 5.0 | 4.8 | 5.6 | 5.8 |
| HAM10K Loss | 0.3 | 0.2 | 0.3 | 0.3 | 0.3 | 0.2 | 0.2 | 0.2 | 0.2 | 0.2 | 0.6 | 0.2 | 0.2 | 0.2 | 0.2 | 0.2 | 0.2 | 0.2 | 0.2 | 0.2 | 0.2 |
| **Task Mean** | 57.0 | 56.7 | 56.5 | 56.7 | 56.2 | 56.4 | 56.2 | 55.5 | 53.6 | 55.3 | 56.0 | 45.0 | 46.0 | 39.4 | 37.4 | 37.0 | 36.7 | 40.6 | 40.8 | 37.7 | 36.2 |
| **Medical Seg** | | | | | | | | | | | | | | | | | | | | | |
| ACDC Dice Score | 0.6 | 0.5 | 0.5 | 0.5 | 0.4 | 0.5 | 0.5 | 0.5 | 0.4 | 0.4 | 0.6 | 0.4 | 0.2 | 0.5 | 0.2 | 0.2 | 0.3 | 0.3 | 0.2 | 0.3 | 0.3 |
| ACDC Mean Acc@ | 86.3 | 85.8 | 83.4 | 78.5 | 75.5 | 78.0 | 76.9 | 79.4 | 74.0 | 93.4 | 94.1 | 71.7 | 67.6 | 76.0 | 46.7 | 53.7 | 54.5 | 60.3 | 56.1 | 50.8 | 50.9 |
| ACDC Overall Acc@ | 86.5 | 86.2 | 83.2 | 78.7 | 75.1 | 78.3 | 77.0 | 79.0 | 73.5 | 93.5 | 94.2 | 71.5 | 67.5 | 76.0 | 47.2 | 53.4 | 54.2 | 60.3 | 55.5 | 51.4 | 51.4 |
| ACDC mIoU | 57.9 | 57.0 | 57.4 | 53.1 | 50.2 | 53.0 | 47.7 | 54.3 | 50.1 | 66.9 | 67.2 | 47.5 | 47.9 | 50.8 | 27.6 | 30.4 | 35.6 | 35.1 | 32.1 | 24.3 | 26.9 |
| **Task Mean** | 57.8 | 57.3 | 56.1 | 52.7 | 50.3 | 52.4 | 50.5 | 53.3 | 49.5 | 63.6 | 64.0 | 47.8 | 45.8 | 50.8 | 30.4 | 34.4 | 36.2 | 39.0 | 36.0 | 31.7 | 32.4 |
| **Img to Txt ZS** | | | | | | | | | | | | | | | | | | | | | |
| Flickr30K Img2Txt Acc@1 | 6.3 | 6.3 | 7.0 | 5.9 | 5.6 | 6.8 | 5.9 | 5.2 | 4.5 | 4.1 | 3.7 | 4.7 | 4.2 | 1.6 | 1.8 | 2.0 | 1.9 | 2.0 | 1.9 | 1.8 | 1.6 |
| Flickr30K Img2Txt Acc@5 | 20.9 | 21.3 | 21.0 | 20.0 | 19.3 | 22.1 | 20.4 | 18.8 | 18.0 | 16.0 | 16.1 | 16.9 | 15.5 | 8.1 | 8.6 | 8.4 | 8.9 | 9.1 | 9.1 | 8.5 | 8.4 |
| Flickr30K Img2Txt Loss | 3.8 | 3.8 | 3.8 | 3.8 | 3.9 | 3.7 | 3.8 | 3.9 | 3.9 | 3.9 | 3.9 | 4.0 | 4.0 | 4.2 | 4.1 | 4.1 | 4.1 | 4.1 | 4.1 | 4.2 | 4.1 |
| Flickr30K Txt2Img Acc@1 | 5.7 | 5.9 | 6.0 | 5.3 | 5.1 | 6.5 | 6.0 | 5.1 | 5.0 | 3.8 | 4.0 | 4.2 | 3.9 | 1.7 | 1.8 | 2.0 | 2.2 | 2.3 | 1.9 | 1.7 | 1.6 |
| Flickr30K Txt2Img Acc@5 | 20.9 | 22.1 | 21.6 | 20.8 | 20.0 | 23.0 | 21.0 | 19.8 | 18.9 | 16.5 | 17.3 | 17.1 | 15.5 | 7.8 | 8.9 | 8.4 | 9.2 | 9.4 | 9.5 | 8.8 | 8.3 |
| Flickr30K Txt2Img Loss | 3.8 | 3.8 | 3.8 | 3.9 | 3.9 | 3.8 | 3.8 | 3.9 | 3.9 | 3.9 | 4.0 | 4.0 | 4.0 | 4.2 | 4.2 | 4.2 | 4.1 | 4.1 | 4.1 | 4.2 | 4.2 |
| NYCC Img2Txt Acc@5 | 21.4 | 21.4 | 22.0 | 20.0 | 21.2 | 22.1 | 21.4 | 20.0 | 17.8 | 17.1 | 17.0 | 15.9 | 15.8 | 7.9 | 8.7 | 8.9 | 8.7 | 9.5 | 8.9 | 8.5 | 7.9 |
| NYCC Img2Txt Loss | 3.8 | 3.8 | 3.8 | 3.8 | 3.8 | 3.8 | 3.8 | 3.8 | 3.9 | 3.9 | 3.9 | 4.0 | 4.0 | 4.2 | 4.1 | 4.1 | 4.1 | 4.1 | 4.1 | 4.1 | 4.2 |
| NYCC Img2Txt | 6.9 | 6.6 | 6.9 | 5.8 | 6.5 | 6.9 | 6.4 | 6.0 | 4.7 | 4.9 | 4.1 | 4.6 | 4.2 | 1.6 | 2.1 | 1.8 | 1.9 | 2.1 | 2.0 | 1.6 | 1.6 |
| NYCC Loss | 3.8 | 3.8 | 3.8 | 3.8 | 3.8 | 3.8 | 3.8 | 3.8 | 3.9 | 3.9 | 3.9 | 4.0 | 4.0 | 4.9 | 4.1 | 4.1 | 4.1 | 4.1 | 4.1 | 4.2 | 4.2 |
| NYCC Txt2Img Acc@5 | 21.9 | 21.6 | 22.5 | 20.2 | 21.9 | 21.9 | 22.7 | 20.7 | 18.4 | 17.3 | 17.4 | 16.0 | 15.3 | 7.9 | 9.4 | 8.3 | 9.4 | 9.9 | 8.9 | 8.9 | 7.9 |
| NYCC Txt2Img Loss | 3.8 | 3.8 | 3.8 | 3.8 | 3.8 | 3.8 | 3.8 | 3.9 | 3.9 | 3.9 | 3.9 | 4.0 | 4.0 | 5.5 | 4.1 | 4.1 | 4.1 | 4.1 | 4.1 | 4.2 | 4.2 |
| NYCC Txt2Img | 6.1 | 5.9 | 6.4 | 5.5 | 6.0 | 6.2 | 6.4 | 5.8 | 4.8 | 4.3 | 4.1 | 3.9 | 3.7 | 1.6 | 2.0 | 1.7 | 2.0 | 2.4 | 1.9 | 1.8 | 1.6 |
| Winoground Img2Txt Loss | 0.7 | 0.7 | 0.7 | 0.7 | 0.7 | 0.7 | 0.7 | 0.7 | 0.7 | 0.7 | 0.7 | 0.7 | 0.7 | 0.7 | 0.7 | 0.7 | 0.7 | 0.7 | 0.7 | 0.7 | 0.7 |
| Winoground Img2Txt | 51.0 | 53.4 | 59.5 | 49.7 | 50.0 | 50.3 | 49.5 | 43.5 | 53.8 | 61.9 | 50.0 | 48.9 | 47.3 | 43.9 | 50.0 | 41.3 | 50.0 | 53.2 | 49.6 | 50.1 | 50.4 |
| Winoground Txt2Img Loss | 0.7 | 0.7 | 0.7 | 0.7 | 0.7 | 0.7 | 0.7 | 0.7 | 0.7 | 0.7 | 0.7 | 0.7 | 0.7 | 0.7 | 0.7 | 0.7 | 0.7 | 0.7 | 0.7 | 0.7 | 0.7 |
| Winoground Txt2Img | 50.0 | 55.2 | 56.2 | 53.1 | 50.0 | 55.5 | 48.3 | 54.2 | 48.6 | 54.8 | 50.0 | 49.6 | 52.4 | 52.8 | 50.0 | 54.2 | 51.8 | 52.2 | 51.8 | 48.8 | 52.1 |
| **Task Mean** | 21.1 | 22.0 | 22.9 | 20.6 | 20.6 | 22.1 | 20.8 | 19.9 | 19.4 | 20.0 | 18.4 | 18.2 | 17.8 | 13.5 | 14.3 | 13.7 | 14.6 | 15.2 | 14.6 | 14.1 | 14.1 |
| **Video Class** | | | | | | | | | | | | | | | | | | | | | |
| HMDB-51 Acc@1 | 52.5 | 40.7 | 40.6 | 32.2 | 39.3 | 24.9 | 27.4 | 32.8 | 33.1 | 5.6 | 11.5 | 1.8 | 2.1 | 3.8 | 8.3 | 7.9 | 6.1 | 5.4 | 6.4 | 7.5 | 4.0 |
| HMDB-51 Acc@5 | 81.4 | 70.0 | 70.5 | 60.9 | 68.6 | 54.2 | 58.5 | 59.8 | 63.8 | 23.0 | 28.8 | 10.4 | 10.2 | 13.6 | 26.4 | 25.3 | 17.8 | 23.6 | 24.4 | 24.9 | 15.6 |
| HMDB-51 Loss | 2.1 | 2.8 | 3.1 | 3.4 | 2.7 | 3.8 | 3.3 | 3.1 | 3.0 | 4.7 | 4.4 | 4.7 | 4.1 | 3.9 | 4.2 | 4.3 | 4.4 | 3.7 | 3.8 | 3.7 | 3.9 |

| | | | | | | | | | | | | | | | | | | | | | |
|---|---|---|---|---|---|---|---|---|---|---|---|---|---|---|---|---|---|---|---|---|---|
| Kinetics Acc@1 | 48.8 | 44.2 | **51.4** | 43.7 | 40.3 | 44.6 | 33.2 | 36.4 | 25.8 | 2.7 | 1.0 | 0.2 | 0.3 | 0.4 | 2.0 | 1.6 | 1.0 | 0.5 | 0.3 | 0.3 | 0.3 |
| Kinetics Acc@5 | 75.5 | 70.9 | **77.9** | 70.7 | 67.6 | 71.7 | 59.9 | 63.0 | 51.8 | 9.7 | 4.3 | 1.3 | 1.4 | 1.7 | 7.0 | 6.5 | 3.5 | 2.2 | 1.3 | 1.3 | 1.3 |
| Kinetics Loss | 2.4 | 2.6 | **2.1** | 2.5 | 2.7 | 2.5 | 3.2 | 3.0 | 3.5 | 5.5 | 6.1 | 6.1 | 6.1 | 6.1 | 5.7 | 5.8 | 6.0 | 6.1 | 6.1 | 6.1 | 6.1 |
| UCF-101 Acc@1 | **84.4** | 75.1 | 69.9 | 63.2 | 75.0 | 63.4 | 58.8 | 66.6 | 48.7 | 19.7 | 11.1 | 2.8 | 0.8 | 2.1 | 15.2 | 13.3 | 6.6 | 8.7 | 6.5 | 7.0 | 2.7 |
| UCF-101 Acc@5 | **95.4** | 92.5 | 89.1 | 82.3 | 91.6 | 86.2 | 81.7 | 86.3 | 75.3 | 42.2 | 28.9 | 8.5 | 5.0 | 8.2 | 35.5 | 33.8 | 17.9 | 25.2 | 23.1 | 20.2 | 11.2 |
| UCF-101 Loss | **0.6** | 1.0 | 1.3 | 1.7 | 1.0 | 1.5 | 1.7 | 1.4 | 2.3 | 4.3 | 5.0 | 4.8 | 4.7 | 4.6 | 3.7 | 3.8 | 4.5 | 4.0 | 4.2 | 4.2 | 4.5 |
| **Task Mean** | **73.0** | 65.6 | 66.6 | 58.8 | 63.7 | 57.5 | 53.3 | 57.5 | 49.8 | 17.2 | 14.3 | 4.2 | 3.3 | 5.0 | 15.7 | 14.7 | 8.8 | 10.9 | 10.3 | 10.2 | 5.8 |
| **Video Reg** | | | | | | | | | | | | | | | | | | | | | |
| IWildCam MAE Score | 1.3 | 1.4 | 1.3 | 1.4 | 1.4 | 1.6 | 1.4 | 1.5 | 1.6 | 2.0 | 1.9 | 1.9 | **2.6** | 2.1 | 1.8 | 1.8 | 1.9 | 1.8 | 2.2 | 1.8 | 2.1 |
| IWildCam MSE Loss | **3.7** | 4.4 | 4.0 | 4.0 | 4.1 | 5.4 | 4.3 | 5.0 | 5.9 | 7.1 | 6.5 | 6.2 | 12.5 | 8.5 | 5.1 | 6.3 | 6.0 | 6.2 | 8.6 | 6.4 | 8.4 |
| **Task Mean** | 1.3 | 1.4 | 1.3 | 1.4 | 1.4 | 1.6 | 1.4 | 1.5 | 1.6 | 2.0 | 1.9 | 1.9 | **2.6** | 2.1 | 1.8 | 1.8 | 1.9 | 1.8 | 2.2 | 1.8 | 2.1 |
| **GATE** | | | | | | | | | | | | | | | | | | | | | |
| Full GATE Mean | **69.0** | 66.8 | 66.8 | 64.6 | 64.3 | 63.4 | 62.1 | 62.2 | 58.5 | 56.3 | 54.4 | 48.4 | 42.8 | 39.6 | 37.5 | 37.2 | 36.2 | 36.9 | 35.0 | 34.9 | 31.8 |
| Big GATE Mean | **76.6** | 74.5 | 74.4 | 72.8 | 72.0 | 71.9 | 70.6 | 70.0 | 66.8 | 66.7 | 64.8 | 58.5 | 53.1 | 46.8 | 43.8 | 43.4 | 41.9 | 41.5 | 40.9 | 39.8 | 37.1 |
| Base GATE Mean | **68.3** | 65.6 | 65.7 | 62.6 | 63.7 | 60.7 | 60.2 | 60.7 | 58.6 | 55.1 | 53.5 | 48.2 | 42.8 | 38.0 | 36.5 | 36.3 | 35.4 | 36.6 | 34.8 | 34.8 | 30.4 |
| Small GATE Mean | **77.7** | 74.9 | 74.6 | 73.3 | 72.4 | 71.2 | 68.9 | 69.1 | 65.3 | 65.7 | 61.7 | 58.5 | 49.3 | 40.5 | 35.7 | 35.4 | 35.9 | 35.3 | 34.1 | 34.4 | 30.4 |
| Full GATE Rank | **1.0** | 3.0 | 2.0 | 4.0 | 5.0 | 6.0 | 8.0 | 7.0 | 9.0 | 10.0 | 11.0 | 12.0 | 13.0 | 14.0 | 15.0 | 16.0 | 18.0 | 17.0 | 19.0 | 20.0 | 21.0 |
| Big GATE Rank | **1.0** | 2.0 | 3.0 | 4.0 | 5.0 | 6.0 | 7.0 | 8.0 | 9.0 | 10.0 | 11.0 | 12.0 | 13.0 | 14.0 | 15.0 | 16.0 | 17.0 | 18.0 | 19.0 | 20.0 | **21.0** |
| Base GATE Rank | **1.0** | 3.0 | 2.0 | 5.0 | 4.0 | 7.0 | 8.0 | 6.0 | 9.0 | 10.0 | 11.0 | 12.0 | 13.0 | 14.0 | 16.0 | 17.0 | 18.0 | 15.0 | 20.0 | 19.0 | 21.0 |
| Small GATE Rank | **1.0** | 2.0 | 3.0 | 4.0 | 5.0 | 6.0 | 8.0 | 7.0 | 10.0 | 9.0 | 11.0 | 12.0 | 13.0 | 14.0 | 16.0 | 17.0 | 15.0 | 18.0 | 20.0 | 19.0 | **21.0** |

Table 3: Full experiments table: Black/Bold best model, Green second best, Blue third best, and red the worst performing model. Models prefixed with 's' refer to 'from scratch' trained models, rather than pretrained. This table showcases the full set of data we use to evolve GATE using EEVEE.

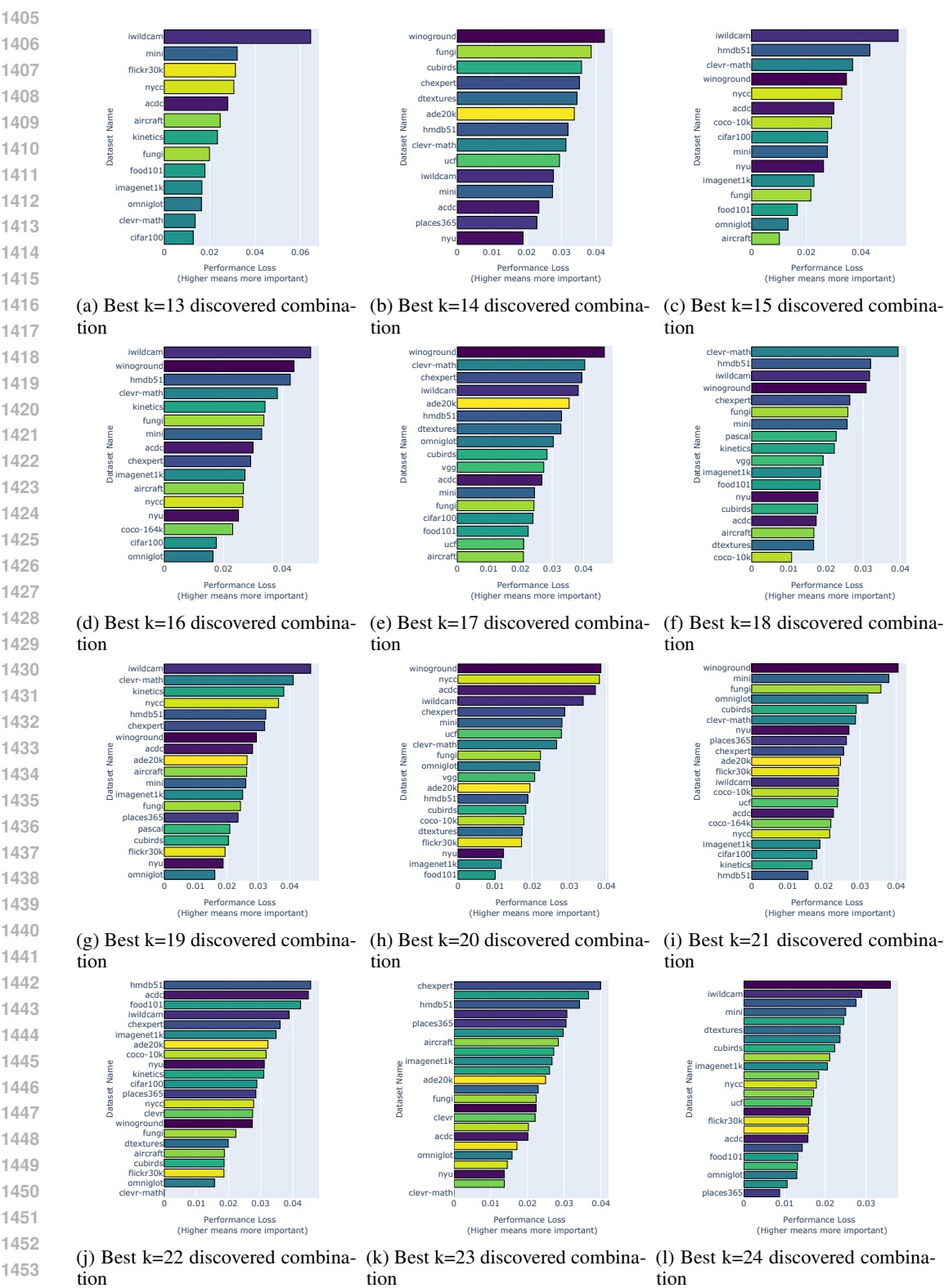

Figure 7: Degradation of predictive power when a given benchmark is removed and the meta-model trained from scratch, for different best combinations in varying $k$.

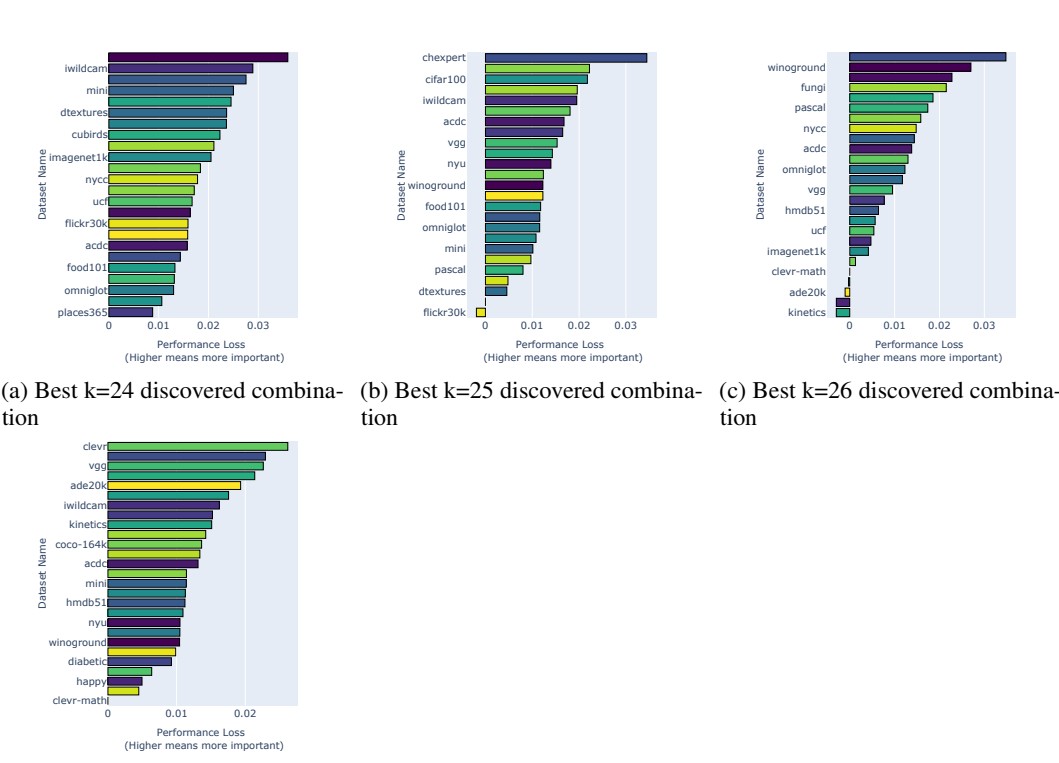

(a) Best k=24 discovered combination

(b) Best k=25 discovered combination

(c) Best k=26 discovered combination

(d) Best k=27 discovered combination

Figure 8: Degradation of predictive power when a given benchmark is removed and the meta-model trained from scratch, for different best combinations in varying $k$.

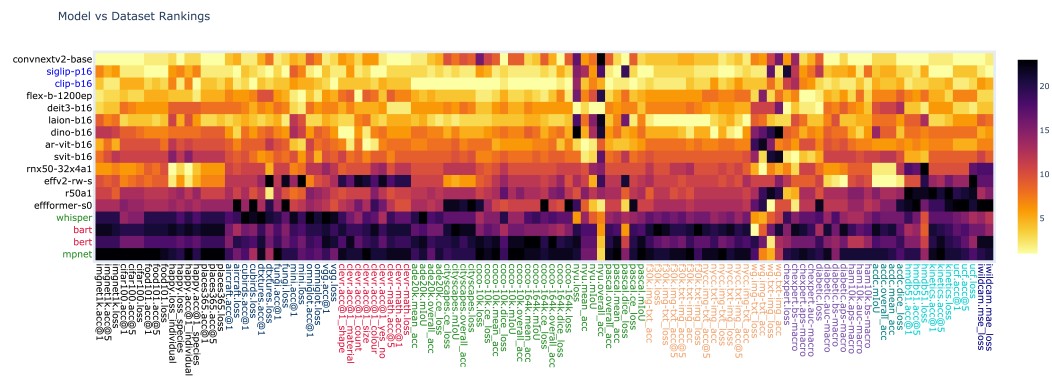

Figure 9: Ranking Heatmap for bigGATE We show how the various models on the y-axis rank on the metrics on the x-axis, where brighter is higher/better rank. From left to right we apply a spearman correlation sorting to capture tasks more similar to imagenet1k more towards the leftmost side, and, dissimilar ones towards the rightmost side. From top to bottom we rank models based on average rank.

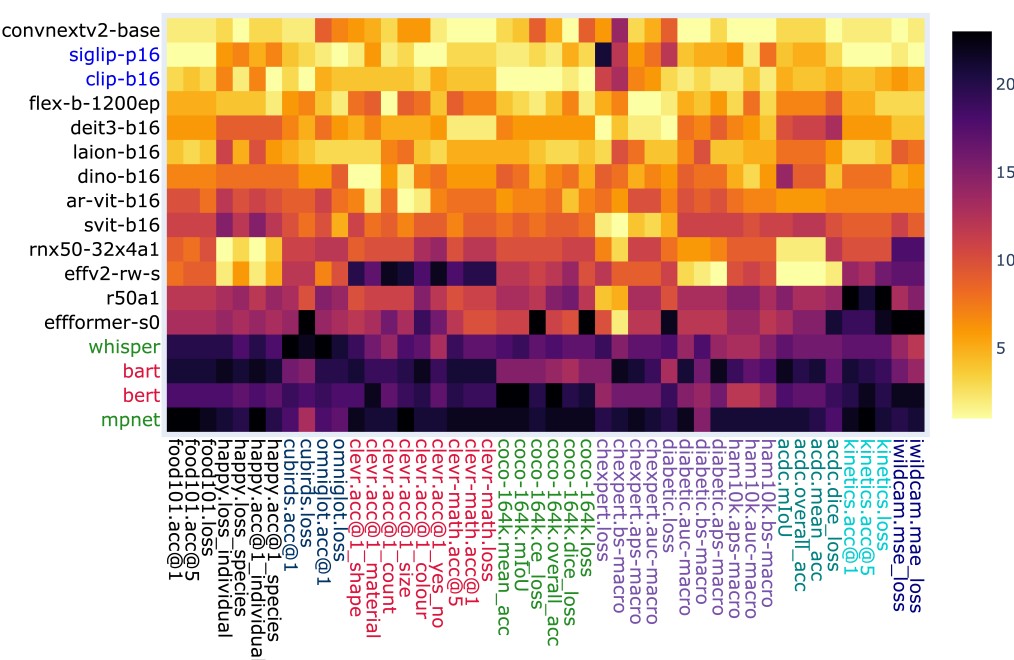

Figure 10: Ranking Heatmap for baseGATE: We show how the various models on the y-axis rank on the metrics on the x-axis, where brighter is higher/better rank. From left to right we apply a spearman correlation sorting to capture tasks more similar to imagenet1k more towards the leftmost side, and, dissimilar ones towards the rightmost side. From top to bottom we rank models based on average rank.

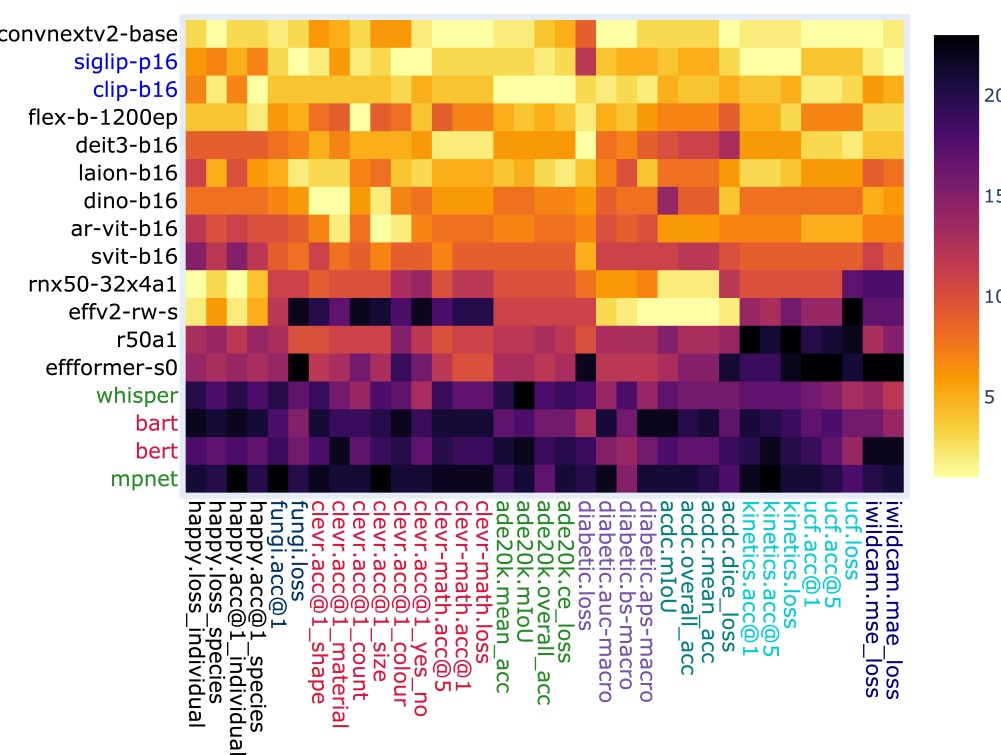

Figure 11: Ranking Heatmap for smallGATE: We show how the various models on the y-axis rank on the metrics on the x-axis, where brighter is higher/better rank. From left to right we apply a spearman correlation sorting to capture tasks more similar to imagenet1k more towards the leftmost side, and, dissimilar ones towards the rightmost side. From top to bottom we rank models based on average rank.

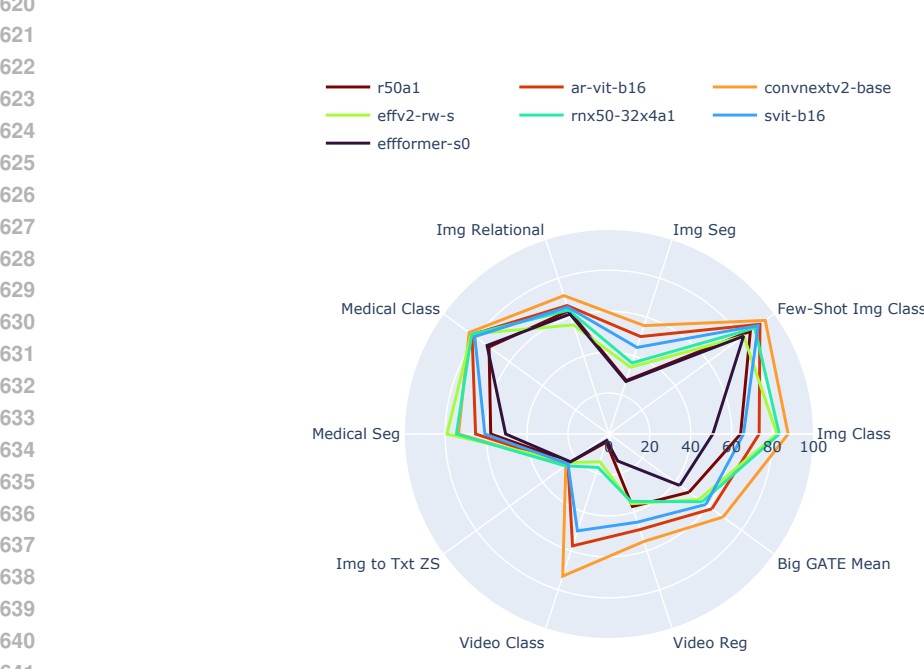

Figure 12: Architecture Variation: Results of keeping the pretraining method the same as ImageNet1k classification and varying the architecture across various key task domains.

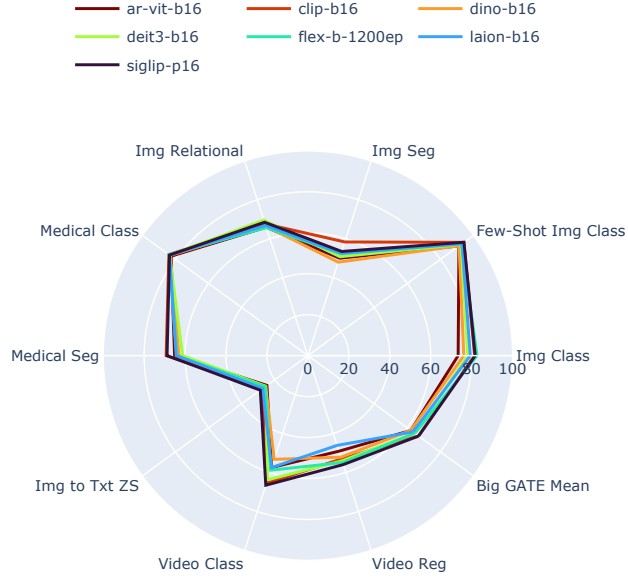

Figure 13: Pretraining Scheme Variation: Results of varying the pretraining method and keeping the architecture as ViT B16 across various key task domains.

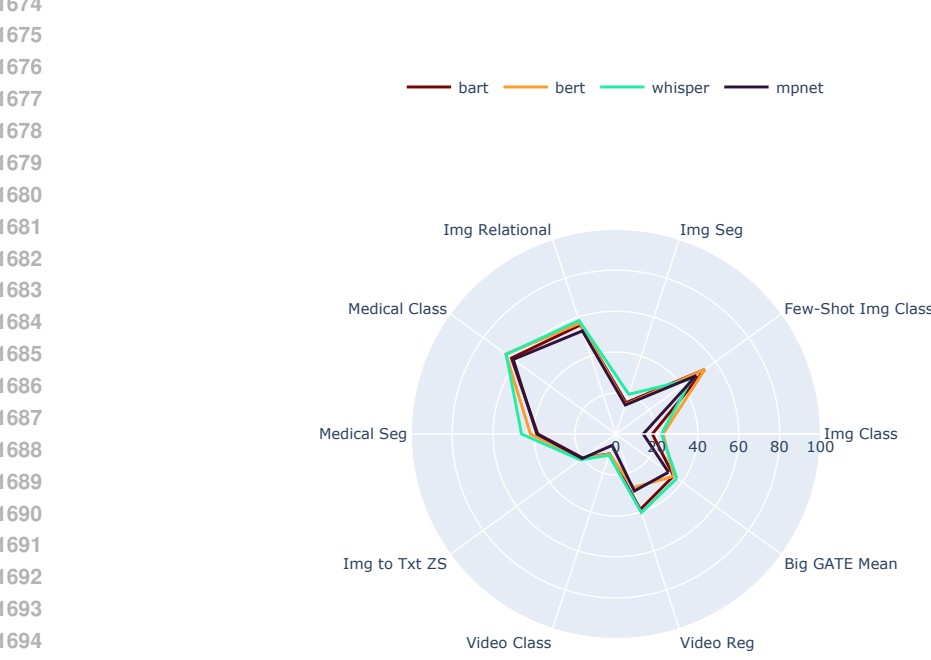

Figure 14: Modality Variation: Results of attempting modality shifting from audio and text to vision tasks.

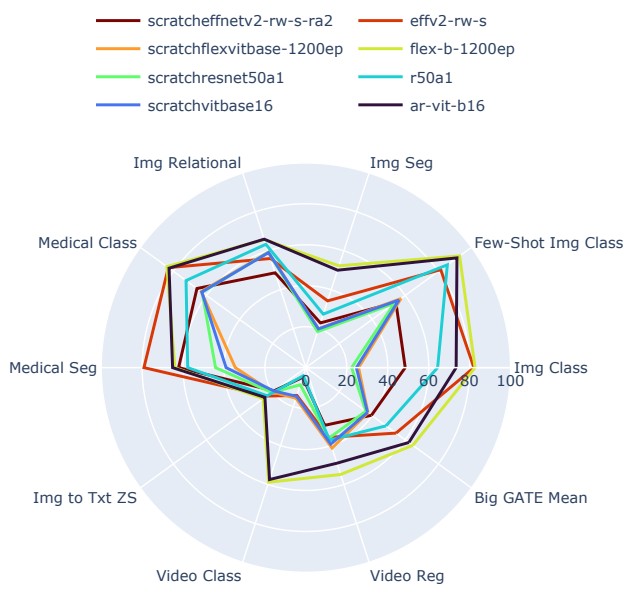

Figure 15: Modality Variation: Results of attempting modality shifting from audio and text to vision tasks.

