# OpenReview forum: "EEVEE and GATE: Finding the right benchmarks and how to run them seamlessly"
_ICLR.cc/2025/Conference — Submitted to ICLR 2025_

### Official Review · Reviewer_S8YR · 2024-10-29

**Soundness:** 2
**Presentation:** 2
**Contribution:** 3
**Rating:** 5
**Confidence:** 3

**Summary:**

The authors propose a new way of benchmarking called EEVEE. They first applied 20 models on 31 datasets of different modalities. EEVEE selects an optimal subset of the dataset to perform the benchmark on and retrieve the score on all the datasets using the meta-learning model. Then, they propose a GATE benchmark allowing the user to download and preprocess the datasets and then train the model on them easily.

**Strengths:**

The authors focus on an essential point of machine learning: how to evaluate models without using too much computation resources. To do that, they proposed a huge work to give access to multiple datasets and models of different modalities to perform benchmarks on them easily. Having a common framework could save a lot of time for numerous researchers. More than that, they propose a meta-learning method to reduce the number of datasets to test in order to have the full benchmark.

**Weaknesses:**

If the paper's ambition is attractive, there are some weaknesses:
- It lacks clarity at different points. The procedure of EEVEE is complex and difficult to follow.
- I understand that the reviewing is blinded, but the authors promise to give an easy framework for the user. It would be nice to have access to the code to criticize that.
- The results part is messy: (1) It would be interesting to have a concrete part explaining the best combination of datasets. Why. And which one is kept to do the  GATE benchmark. Figure 4 is never cited in the text. (2) It could be nice to see if the best models given by the GATE benchmark are the same as with a regular dataset.

**Questions:**

- Can we have one model in train and validation set at the same time? That is what your example says.
- Scoring takes as input M_selected when it says B_initial in the evolution algo?
- Do you try different numbers of mutations? Why do you choose only one?

---

### Official Review · Reviewer_1PYi · 2024-10-31

**Soundness:** 2
**Presentation:** 2
**Contribution:** 2
**Rating:** 3
**Confidence:** 3

**Summary:**

In this paper, the authors tackle the challenge of sub-selecting a larger set of benchmarks into representative subsets, corresponding to different amounts of compute, that are equally as informative for downstream metrics as the larger set would be. The authors have two main contributions. The first, is proposing to run an evolutionary search to find optimal-subsets of algorithms in tandem with using a smaller neural network as a proxy to predict the performance metrics of the subsets. They refer to this learning algorithm as EEVEE, and use it to determine datasets which used together offer high signal-to-GPU hour ratios. The second contribution is that they use EEVEE to construct three sets of benchmarks, which they refer to as the GATE benchmarks, that cater to different compute budgets. For this, they also propose the GATE engine, a software package that standardizes dataset organization and modification.

**Strengths:**

• The set of possible pre-trained models, pre-training schemes, and downstream tasks considered is very comprehensive.

• The subsets that the authors have discovered could serve as useful benchmarks for people who work in the research communities of visual backbones. Standardization of evaluation could be an important contribution.

• It seems like their approach could be scalable as more benchmarks come out over time, as long as there is someone who is able to run their criteria and have the time to make something like the GATE benchmark.

**Weaknesses:**

• The major drawback of the paper is in the central premise that reducing evaluation time would alleviate one of the biggest bottlenecks for researchers, when in fact the majority of time for researchers (not corporations) is spent on training rather than evaluation. Even for folks who they would consider 'GPU Poor', I wonder if the cost of evaluation is really that high (unless the focus is on models that are expensive to evaluate such as closed-source foundation models. For such LLM problems, I could imagine there would be more applicability).

• While the paper does propose a technique for 'distilling' a larger set, I don't think it solves the problem of unifying the sub-selected benchmarks that one would have to do regardless. This was one of the parts of their main motivation and is not resolved. While GATE is a nice standardization for the current set of benchmarks they consider, it's unclear if they propose everyone using GATE as a standard framework, or how it ties into the broader story of making evaluation easier going forward.

• Why is the evolutionary search necessary for the proposed solution to work? In fact, if we only need to construct something like GATE once, why not randomly select subsets and just pick the best of those under different compute budgets (the authors claim that this is intractable which indeed for ALL possible subsets it is, but they do not compare against what would happen if you did do brute-force)? Indeed for considering subsets it seems from the authors own findings in Figure 5 it's difficult anyways with larger subsets to predict performance. It's unclear to me why there needs to be learning at all here, and the authors do not consider other strategies for getting these high-value subsets.

**Questions:**

• From line 206, is there evidence to suggest the subsets that EEVEE selects are Pareto-optimal?

• Are there any contributions that are directly from the way that GATE is constructed? It seems like the 'GATE engine' is one of the major contributions of the paper.

Ultimately, I am left quite confused about how to best rate this paper. I think that the GATE benchmark/suite of standardization could be useful for researchers and that the optimization of benchmark sub-selection could be helpful in determining what models are best in model selection. However, it's unclear from the evidence presented that EEVEE is necessary a good way to do this (over naive baselines which weren't considered) and how GATE allows for future adaptation of other datasets.

---

### Official Review · Reviewer_MbVZ · 2024-11-03

**Soundness:** 3
**Presentation:** 3
**Contribution:** 3
**Rating:** 8
**Confidence:** 3

**Summary:**

This paper proposes EEVEE, a framework for finding the optimal set of benchmarks that accurately reflect a model's quality with minimal evaluation costs. It utilizes evolutionary algorithm to select benchmarks and learns a performance predictor model. To show its effectiveness, the paper applies EEVEE to evaluate a model's out-of-domain adaptation ability. It identifies multiple set of benchmarks with optimal signal-to-GPU-time ratio and suitable for different evaluation budgets. The paper also presents GATE, a benchmarking and software suite to facilitate model evaluation.

**Strengths:**

1. The problem studied---evaluation accuracy and efficiency---is super important, so the proposed work is well-motivated. It would be beneficial to the entire research community if there's a unified way of selecting benchmarks. The idea of framing evaluation process design as a learning problem is also interesting and novel.
2. The contribution of the paper is multifold, including a benchmark identification framework, useful benchmark combinations, software to facilitate evaluation, etc
3. Insights provided by analyzing leading benchmarks and models are beneficial to the research community

**Weaknesses:**

1. It's better to follow the ICLR citation format, e.g., using (Bengio & LeCun, 2007) instead of the number of the references listed in order.
2. Figure 5 is hard to interpret without legend. What do the boxes represent?
3. There's no supplementary material provided. Is it possible to provide a demo or code for the GATE engine (anonymized)?

**Questions:**

See weaknesses

---

### Official Review · Reviewer_Utuj · 2024-11-03

**Soundness:** 1
**Presentation:** 1
**Contribution:** 1
**Rating:** 1
**Confidence:** 4

**Summary:**

This paper introduces EEVEE, a subset selector built using a 2-layer MLP, designed to efficiently select high-signal, low-cost evaluation routines from large datasets. Additionally, the paper presents an extensible and modular framework to support model encoder evaluation experiments, including tasks such as linear probing for image classification with the CLIP vision encoder and handling domain shift scenarios.

**Strengths:**

The research question of identifying the right benchmark is intriguing, practical, and crucial for both the research community and industry, as it enables a fair, efficient, and cost-effective evaluation protocol for model development.

**Weaknesses:**

There are several concerns for the paper, which lead me majorly tend to think the paper is not ready for publication.

1. Scope: The paper has two main contributions: the subset selection module and the customized evaluation framework. From both the writing (particularly the abstract) and the length dedicated to each section, it appears that the latter is emphasized as the primary contribution. However, creating a Python library alone may not offer sufficient insights for researchers, especially considering that the tasks, models, and datasets are already widely used in the community, and numerous other papers provide comprehensive benchmarking results. As a result, the paper leans more toward an engineering contribution, which might align better with a systems-oriented venue rather than ICLR.

2. Overclaim of the Technological Contribution: My main concern is with the subset selection module, named EEVEE in the paper.

(1) Lack of Evidence for Optimal or Pareto-Optimal Selection: The authors do not provide evidence that their proposed method can reliably identify an optimal or Pareto-optimal subset of benchmarks (data and metrics as described in the paper). The methodology relies on an MLP combined with an evolutionary algorithm, neither of which guarantees optimal or Pareto-optimal results, despite the paper’s claims.

(2) Issues with MSE as the Evaluation Metric: Using MSE between the entire set and subset metrics introduces potential biases. For example, model and metric biases could distort results, as different metrics vary in scale and significance (e.g., some metrics are favorable when >0.5, whereas others, like random guessing in binary classification, are not). Additionally, there are established works in dataset selection and dataset knowledge distillation that the paper does not reference or compare.

(3) Lack of Comprehensive Domain and Modality Support: Although the paper claims to support multiple domains and modalities, no language tasks are included in the evolutionary process. Furthermore, the submission lacks demos or supplementary materials to help reviewers assess the framework’s functionalities.

3. The writing needs improvement for clarity and readability, as it is currently challenging to follow. Some key terms, like “predictive power,” lack proper definition. The authors should clarify whose predictive power is being referenced and how it is measured (for instance, the MSE between which specific components). Additionally, the writing is overly wordy and not well-organized. In sections like the abstract and introduction, listing model and dataset names is often unnecessary when citations alone would suffice. Finally, the related work section is outdated, as it primarily compares with older benchmarks, like ImageNet, while many new benchmarks have emerged recently.

**Questions:**

My questions mainly comes from the definition of the terms used in the paper:

1. In the statement, “using EEVEE we quantified the predictive power of each benchmark on its own,” what exactly is meant by “predictive power” of a benchmark? Predictive power typically refers to a model’s capability rather than that of a benchmark (dataset + task). Are you referring to the average performance metrics of models on each benchmark?

2. In the phrase, “Winoground are the least predictive tasks,” does this imply that the dataset is challenging for models to learn and achieve good performance? If so, could you clarify this definition of “least predictive”?

3. In Figure 5, there are 26 “number of contexts,” whereas the text mentions “top 50 combination candidates.” Does “contexts” refer to “candidates,” or are they different concepts?

4. In the statement, “our benchmark produces fine-grained information,” what is meant by “fine-grained information”? Are you referring to gradients or some other specific data?

5. On page 4, the term “rich evaluation signals” is used. Could you clarify what is meant by “signals” in this context? I noticed only evaluation metrics are provided, so in what way are these metrics considered “rich” compared to other works?

---

### Meta-Review · Area_Chair_UCcP · 2024-12-20

**Metareview:**

This paper presents a method to distill large sets of benchmark evaluation datasets into smaller, but equally informative, datasets to reduce the evaluation computational costs. They use this to create subset benchmarks that allow for different compute budgets and present a software framework that standardizes the evaluation workflow on these.

Strengths: Many models and tasks evaluated, standardization of evaluation workflow can be an important contribution
Weaknesses: It is unclear how essential the learning method for finding optimal subsets is compared to other approaches, as well as whether it truly achieves optimality, it is difficult to evaluate the software framework (a main contribution) given the lack of code/software details, and the paper presentation is dense making it difficult to follow the ideas presented.

The paper could benefit from a re-organization to simpler and more precise text with a motivating discussion on why evaluation compute budget optimization is critical (as opposed to training compute budgets that are typically much larger), experimental evidence to suggest the necessity of the learning method and comparisons to other types of methods to achieve the same goal, and a more detailed description of the software framework. Adding language tasks could also help from making the evaluation more comprehensive.

**Additional Comments On Reviewer Discussion:**

Clarity of methods, experimental evidence to show the necessity of the learning, and more details on the software were the main concerns raised. While reviewer MbVZ was positive about the contributions (that were also acknowledged by other reviewers), the concerns raised from other reviewers are still significant.

There was no author rebuttal to address the concerns.

---

### Decision · Program_Chairs · 2025-01-22

Reject